# Representativeness assessment of the pan-Arctic eddy-covariance site network, and optimized future enhancements

Martijn M. T. A. Pallandt[1], Jitendra Kumar[2], Marguerite Mauritz[3], Edward A. G. Schuur[4], Anna-Maria Virkkala[5], Gerardo Celis[6], Forrest M. Hoffman[7], Mathias Göckede[1]

[1]Department of Biogeochemical Signals, Max Planck Institute for Biogeochemistry, Jena, 07745, GER

[2]Environmental Sciences Division, Oak Ridge National Laboratory, Oak Ridge, TN 37831, USA

[3]Department of Biological Sciences, The University of Texas at El Paso, El Paso, TX 79902, USA

[4]Center for Ecosystem Science and Society, and Department of Biological Sciences, Northern Arizona University, Flagstaff, AZ 86011, USA

[5]Woodwell Climate Research Center, Falmouth, MA 02540, USA

[6]Agronomy Department, University of Florida, Gainesville, FL 32601, USA

[7]Computer Science and Engineering Division, Oak Ridge National Laboratory, Oak Ridge, TN 37831, USA

*Correspondence to*: Martijn M. T. A. Pallandt (mpall@bgc-jena.mpg.de)

**Abstract.** Large changes in the Arctic carbon balance are expected as warming linked to climate change threatens to destabilize ancient permafrost carbon stocks. The eddy covariance (EC) method is an established technique to quantify net losses and gains of carbon between the biosphere and atmosphere at high spatio-temporal resolution. Over the past decades, a growing network of terrestrial EC tower sites has been established across the Arctic, but a comprehensive assessment of the network's representativeness within the heterogeneous Arctic region is still lacking. This creates additional uncertainties when integrating flux data across sites, for example when upscaling fluxes to constrain pan-Arctic carbon budgets, and changes therein.

This study provides an inventory of Arctic (here $>= 60^{O}N$) EC sites, which has also been made available online (https://cosima.nceas.ucsb.edu/carbon-flux-sites/). Our database currently comprises 120 EC sites, but only 83 are listed as active, and just 25 of these active sites remain operational throughout the winter. To map the representativeness of this EC network, we evaluated the similarity between environmental conditions observed at the tower locations and those within the larger Arctic study domain based on 18 bioclimatic and edaphic variables. This allows us to assess a general level of similarity between ecosystem conditions within the domain, while not necessarily reflecting changes in greenhouse gas flux rates directly. We define two metrics based on this representativeness score, one that measures whether a location is represented by an EC tower with similar characteristics (ER1), and a second for which we assess if a minimum level of

representation for statistically rigorous extrapolation is met (ER4). We find that while half of the domain is represented by at least one tower, only a third has enough towers in similar locations to allow reliable extrapolation. When we consider methane measurements or year-round (including wintertime) measurements, the values drop to about one fifth and one tenth of the domain respectively. With the majority of sites located in Fennoscandia and Alaska, these regions were assigned the highest level of network representativeness, while large parts of Siberia and patches of Canada were classified as under-represented. Across the Arctic, particularly mountainous regions were poorly represented by the current EC observation network.

We tested three different strategies to identify new site locations, or upgrades of existing sites, that optimally enhance the representativeness of the current EC network. While 15 new sites can improve the representativeness of the pan-Arctic network by 20 percent, upgrading as few as 10 existing sites to capture methane fluxes, or remain active during wintertime, can improve their respective ER1 network coverage by 28 to 33 percent. This targeted network improvement could be shown to be clearly superior to an unguided selection of new sites, therefore leading to substantial improvements in network coverage based on relatively small investments.

**Section 1: Introduction**

Because of the vastness, inaccessibility and extreme climate of the Arctic zone, research in this region is a complex endeavor. There are large pools of soil organic carbon in the Arctic (Yu, 2012; Hugelius et al., 2014; Schuur et al., 2013; Strauss et al., 2017; Nichols and Peteet, 2019; Mishra et al., 2021) that have accumulated over the past millennia, which are at increased risk of thawing linked to climate change and its associated Arctic amplification (Schuur et al., 2008; Serreze and Barry, 2011; IPCC, 2014; Schuur et al., 2015; Meredith et al., 2019; Hugelius et al., 2020). With limited insights into current Arctic carbon cycle processes, it is difficult to determine trends and changes in Arctic carbon budgets (Belshe et al., 2013; McGuire et al., 2012; Oechel et al., 2014; Pörtner et al., 2019; Bruhwiler et al., 2021). Therefore, our ability to establish quantitative links between climate change and carbon processes, and to forecast future carbon cycle processes, is severely limited, especially when regarding wintertime fluxes (Zimov et al., 1996; Wille et al., 2008; Euskirchen et al., 2012; Marushchak et al., 2013; Lüers et al., 2014; Oechel et al., 2014; Natali et al., 2019).

Eddy covariance (EC) is a widely used method to measure ecosystem-scale greenhouse gas fluxes (Baldocchi, 2003; Sulkava et al., 2011; Pastorello et al., 2020). The method is non-destructive, and allows continuous monitoring of surface-atmosphere exchange fluxes at high temporal frequency (Baldocchi et al., 1988; Lee et al., 2005; Burba and Anderson, 2010; Aubinet et al., 2012). Despite the difficulties listed above, many EC sites that measure greenhouse gases fluxes have been established in the Arctic (Kutzbach et al., 2007; Dolman et al., 2012; Ueyama et al., 2013; Zona et al., 2014; Emmerton et al., 2016; Zona et al., 2016; Parmentier et al., 2017), which for this study we consider as the region north of 60 degrees latitude. Most of these sites are affiliated with global and regional EC flux networks (e.g. Fluxnet, AmeriFlux, Asiaflux, Integrated Carbon Observation System) facilitating multi-site syntheses. However, to date there is no such network that specifically lists all the

sites in the Arctic. Moreover, beyond the fact that metadata information for specific sites sometimes differs between these networks, some sites are simply not listed in any of them, which makes it difficult for scientists working in this domain to gain a clear overview of all available EC data.

Knowing the current and past spatiotemporal distribution of EC sites is not enough to fully understand to which degree this network represents the Arctic domain. The reason for this is that EC towers have a field of view that typically does not extend further than a kilometer from the tower, often less (Leclerc and Thurtell, 1990; Horst and Weil, 1992; Schmid, 1994, 2002; Vesala et al., 2008). Accordingly, with currently about 120 terrestrial EC towers situated within the Arctic domain, only a very small fraction of the region gets directly observed, while most of its expanse remains unsampled. Larger
footprints would not solve this problem, as the greater heterogeneity would still be hard to capture. Meteorology, vegetation, above/below ground conditions, and topography are critical drivers of hydrological and biogeochemical processes at landscape scale and of greenhouse gas (GHG) fluxes, and their variability across the Arctic therefore also causes variability in flux rates. For upscaling purposes (i.e., when fluxes are predicted over larger areas), typically a tower is held as representative for the ecosystem and the region where it is stationed (Desai, 2010; Jung et al., 2011; Xiao et al., 2012; Chu et
al., 2021); however, except when using a very coarse classification of ecosystem types, the existing EC network still cannot cover all ecosystems across the Arctic, and a coarser classification would increase heterogeneity within the ecosystem and reduce the representation within the ecosystem. Still, a number of published studies have successfully demonstrated the effectiveness of using meteorological and environmental variables as explanatory variables for estimating GHG fluxes at regional to global scales (e.g. Jung et al., 2020; Knox et al., 2019).

There have been several studies that aim at evaluating the spatial coverage of regional EC sites (Sulkava et al., 2011; Hoffman et al., 2013; Chu et al., 2021; Villarreal and Vargas, 2021). For the high precision concentration atmospheric tall tower networks, which can be utilized to integrate fluxes on regional scales through their large footprints, similar studies have been performed (Shiga et al., 2013; Ziehn et al., 2014; Kountouris et al., 2018), though none of these focused on the Arctic. Even though the patchiness of Arctic field sampling locations has received more attention lately (Metcalfe et al.,
2018; Virkkala et al., 2019), so far only the distribution of Arctic chamber network has been extensively summarized (Virkkala et al., 2018). Thus, overall we find no detailed analysis of the Arctic EC network. Especially the pronounced spatial variability in Arctic ecosystem characteristics across scales make this evaluation more difficult (Lara et al., 2020; Tuovinen et al., 2019; Virkkala et al., 2021), but at the same time highly important.

Building on a study by Hoffman et al. (2013) that presented an analysis of the Alaskan EC network, in this study we will
provide a first in-depth evaluation of the current and past pan-Arctic EC flux observation infrastructure. Our method uses quantitative multivariate clustering, which has many uses from creating maps of geological regions (Harff and Davis, 1990), to watershed delineation (Hessburg et al., 2000) and ecoregion classification (Zhou et al., 2003). Hargrove and Hoffman (2004) give an extensive overview of these applications, which are based on the concept of mapping normalized ecosystem variables such as topography, precipitation and temperature in an n-dimensional data space, using one axis for each variable.
The closer two points are in this variable space, the more alike they are, and the more likely they are to be classified as

belonging to the same ecoregion when clustered by a k-means algorithm. Thus, the distance can be interpreted as a metric of variability. Aiming at assessing the representativeness of the EC network in the US, Hargrove and Hoffman (2004) then calculated the distances between each constructed ecoregion without an EC site to the closest ecoregion with an EC site. Hoffman (2013) later extended this method to map the Alaska EC network. Instead of aggregating the distances between

ecoregions, they calculated the distance between each pixel in the map and the closest EC site. This approach thus preserves the fine scale variability that is lost when aggregating to the ecoregion level. In our implementation we will also perform this analysis on an individual pixel scale.

Our analysis aims to quantify representativeness in the pan-Arctic domain based on this similarity in key ecosystem characteristics of any location in our domain to those of the EC sites. We further use the analysis by Hill et al. (2017) on the

statistical power of EC systems to put these representativeness measures into perspective regarding the general potential to upscale fluxes from sparse EC networks. Moreover, we use the results from the representativeness analyses to identify the most suitable locations for new observation sites, and upgrades to existing infrastructure that would optimally enhance the performance of the Arctic EC network as a whole. Finally, this manuscript and its corresponding online tool aim at providing an easily accessible source of information on Arctic flux monitoring infrastructure for scientists working on the carbon cycle.

**Section 2: Methods**

**2.1 Assessment of flux site infrastructure**

To properly assess the extent of the Arctic EC network, a comprehensive inventory is required of all eddy-covariance flux sites within the domain. To achieve this goal, as a first step we combined metadata (i.a. PI contact information, site name and ID, species sampled, sampling activity, auxiliary measurements) of those sites listed within these established flux networks:

Fluxnet (https://fluxnet.fluxdata.org), Ameriflux (https://ameriflux.lbl.gov/sites/site-search), the European Fluxes Database Cluster (http://www.europe-fluxdata.eu/home/sites-list), ICOS (https://www.icos-cp.eu), and Asiaflux (https://db.cger.nies.go.jp/asiafluxdb). The initial search for EC sites was restricted to those located north of 60 degrees latitude. Even though this publicly available information already covered a large part of the final site list, we discovered a few limitations with these datasets. First, in some cases when a site would appear in several databases, metadata was not

always consistent between them. Second, often some part of the metadata fields was missing, especially detailed information on temporal coverage. Here, generally only start and, if applicable, end times were mentioned, while no information was provided on the seasonal discontinuation of operation that is important particularly at Arctic sites, many of which are only operated during the growing season. Third, a considerable number of sites were not listed in any of the flux networks listed above.

To acquire more comprehensive site-level metadata, we conducted an online survey among principal investigators (PIs, contacted through personal network and Fluxnet newsletter) of flux sites in the Arctic. In addition to confirming basic

information such as exact location, contact information, and, where applicable, references that describe site operations in detail, we specifically asked for the following items:

- Detailed times of operation (on a monthly scale), broken up by $CO_2$ and $CH_4$ fluxes
- List of gas species measured
- Details on eddy-covariance instrumentation (e.g. types of sonic anemometer and gas analyser)
- Details on auxiliary measurements, for example snow depth and precipitation, including power supply
- Mode of data availability (e.g. open, password restricted, upon request).

At the time of writing, we have received 66 responses to our metadata request from site PIs. For all sites for which new data was provided by PIs, in our final site list we used the more recent information from our survey to replace existing information from the databases. We contacted PIs and flux networks in case of conflicting information.

An overview of the eddy covariance flux network that our list comprises will be given in the results section below. To make this information accessible to the Arctic research community, we created an online mapping tool hosted by the Arctic Data Centre of the National Centre for Ecological Analysis and Synthesis (https://www.nceas.ucsb.edu/arctic-data-center). This tool combines several datasets: the EC site set of this paper, a chamber flux set and atmospheric tower set. It also comprises several sites $>50^O$N to encompass the majority of high-latitude permafrost regions, and is accessible at http://cosima.nceas.ucsb.edu/carbon-flux-sites as an easy-to-use web interface that allows the user to identify data availability within certain regions, timeframes, or biome types. The main tool consists of three elements: The central interface holds maps in several layers where the location of the sites is shown, and basic information can be retrieved in popup windows. Furthermore, a panel allows selections of sites based on type, location, activity and duration of observations, while a table at the bottom contains detailed information on all selected sites and, if available, direct links to the actual data are provided. Lists of selected sites for a given search can be downloaded as csv files.

## 2.2 Representativeness assessment

We applied a method described by Hoffman et. al., (2013) and Hargrove et. al., (2003) to calculate a unit-less relative measure of dissimilarity between a location containing an observation site and any other location of interest within the gridded study domain based on underlying datasets that describe the environmental characteristics of a particular site. Dissimilarity between two locations is calculated as Euclidean distance in standardized $n$-dimensional state space. The resulting representativeness score has a minimum 0 (best score, indicating no difference) and a virtual infinite maximum. To improve the comparison between different scenario simulations, all values are normalized to a range between 0 and 1. Due to infrequent but very large positive outliers, the network-wide distribution of representativeness scores is very skewed, and 95% of the normalized values fall within the range 0 - 0.03. Accordingly, for central, aggregated values we report the median.

This method quantifies the similarity between environmental conditions as a continuously varying measure for every location on the map with respect to the EC site of interest. Inputs to the analysis included the EC sites and their coordinates,

and environmental data describing the conditions of the site and the entire Arctic region. We defined our state space using 18 variables capturing bioclimatic, edaphic and permafrost characteristics of the Arctic landscape (Appendix 1). Variables were chosen to represent the primary environmental conditions that control hydrological, ecological and biogeochemical processes in the broad Arctic landscape and in turn its vegetation characteristics (Natali et al., 2019; Virkkala et al., 2021).

Given that we have an extensive network of EC sites, any location within the study domain will be partially represented by

multiple sites in the network, with varying magnitude of representativeness. To produce a final assessment, for each pixel ($1km^2$) only the single best representativeness value was retained from among all representativeness maps of individual sites to develop a gapless, network-wide representativeness map. Therefore, this final network representativeness map displays on a pixel-by-pixel basis how well each location is linked to its most closely related site, and allows differentiation at high spatial resolution between relatively well-represented and poorly-represented regions within the target domain.

**2.3 Assignment of ecoregions and network-wide representativeness scores**

While representativeness was computed on a pixel-by-pixel basis, we used the concept of ecoregions to aid in landscape scale analysis of the results. The main purpose of the ecoregion is to group the sites into regions of homogeneous characteristics. To maintain consistency in the analysis, ecoregions were generated using an unsupervised k-means clustering approach (Kumar et al., 2011), based on the same 18 variables used for calculating representativeness scores, separating

regions with similar properties in environmental data space and minimizing internal variability. Using this clustering, the Arctic region was divided into 100 sub-regions for our analysis. Our choice to separate the Arctic study domain into k=100 ecoregions is based on the following considerations: First, for a smaller number of k, ecoregions would become excessively large, and therefore increasingly heterogeneous, accordingly they would not represent truly coherent units. Second, separating the domain into a much larger number of k would result in ecoregions so small they would not grant much

improvement over using the raw distance. Accordingly, after conducting sensitivity tests over a range of settings for k (35, 100, 200, 500 and 1000), we selected k=100 as a compromise between ecosystem coherence and representativeness that agrees well with our study objectives.

While statistically delineated and defined by their multivariate environmental characteristics, the resulting regions, however, lack a recognizable label which are desired to interpret and validate the ecoregions. To evaluate the robustness of the

ecoregion assignment, we use the Mapcurves algorithm (Hargrove et al., 2006). Mapcurves calculates a statistical Goodness of Fit (GOF) metric that accounts for spatial match and mismatch over all categories in two maps being compared. We compared clustering based 100 ecoregions with the Circumpolar Arctic Vegetation Map (CAVM) (Raynolds et al., 2019), which translates 100 ecoregions to CAVM categories allowing for easier interpretation of the map while still being able to use the quantitative multivariate characteristics. The key differences between our method and the CAVM rasterized maps is

that for the latter clustering was done on sub regions of the original CAVM map using AVHRR and MODIS (red and infrared channels, and NDVI) as well as elevation data from the Digital Chart of the World. They then aggregated the clustering units to their CAVM vegetation units using a wide range of auxiliary data such as regional vegetation maps, ground-based studies as well as GoogleEarth imagery.

    To facilitate a quantitative assessment of network coverage, and put the results into context, we produced two derived

metrics, subsequently labelled ER1 and ER4. Both include a threshold that allows separation of the study domain into areas that meet a defined requirement and those that do not, based on the representativeness score assigned to each pixel. We calculate these thresholds as the 75% percentile of the distribution of representativeness scores calculated for the All scenario described below, restricted to ecoregions that contain at least 1 site (ER1), or at least 4 sites (ER4), respectively. The ER1 metric represents the domain that is covered similarly to an ecoregion with at least one EC tower, and can thus be

interpreted as the fraction of the study area that the EC network provides basic information on. However, one tower typically does not provide enough information to reliably upscale fluxes to an entire ecoregion. Therefore, we added the ER4 metric to consider a minimum number of 4 towers required to reach a 0.95 statistical power to properly characterize an ecosystems EC fluxes (Hill et al., 2017). The ER4 metric can therefore be interpreted as showing that part of the study area for which the existing EC infrastructure allows upscaling of fluxes with a reasonable confidence. The requirement of 4 towers assumes

relatively flat terrain (Baldocchi, 2003), while hilly or even rougher terrain would require at least 24 towers (Hill et al., 2017); however, since none of our ecoregions encompass this many towers we did not include a metric for this terrain. The chosen cutoff at 75% generally follows studies which concluded that a perfect match between target conditions and observed conditions is unrealistic for EC sites, so that a deviation of 20-25% can still be considered 'homogeneous' (e.g. Göckede et al., 2008). In the presented study, applying this cutoff for each scenario as described below, the derived splitting point of

representativeness values to meet the ER1 metric was calculated as 0.0089, while for the stricter ER4 metric this cutoff was 0.0063.

### 2.4 Network subsets

We evaluated the representativeness of the EC network in the Arctic in a number of different subsets and configurations, with all sites performing $CO_2$ flux measurements, and some additionally monitoring $CH_4$ fluxes as described below:

1.   All sites (*All*): This set contains all sites in our dataset, both past and present, and reflects the network in its most extensive state. This subset serves as the starting point for any recommendations for network extension, since also the currently inactive sites can still contribute data for upscaling activities, model development and synthesis work.

    2.   Active sites (*Active*): This second set of sites includes those that reflect the current network coverage. We selected all sites that were listed as active at the start of 2019.

3.   Long-term operational sites (*5-year*): The third subset comprises sites that have been operational for at least 5 data years since 1993. Data coverage does not necessarily need to be continuous in this context thus both wintertime

gaps as well as discontinuous years are considered here. We included this subset based on the assumption that multiple years of data can account for interannual variability (Chu et al., 2017; Baldocchi, 2020), and therefore provide improved insight into functional relationships between fluxes and environmental conditions.

4.    Wintertime network coverage (*Winter*): In this forth subset, we selected sites that provide data coverage during the Arctic wintertime (October through April, following Natali et al. (2019)). With recent studies demonstrating the importance of wintertime fluxes for year-round flux budgets in the Arctic (Mastepanov et al., 2008; Zona et al., 2016; Natali et al., 2019), information on how well our observational infrastructure can capture these signals across the Arctic domain is crucial.

5.    Sites with methane fluxes (*CH₄*): Even though the total carbon release of methane is much lower compared to $CO_2$ fluxes (McGuire et al., 2012), due to its high global warming potential methane needs to be accounted for when constraining carbon cycle feedbacks with global climate change. This is particularly the case for the large fraction of waterlogged areas throughout the Arctic. Since methane fluxes are far more dependent on microtopography than $CO_2$ fluxes (Peltola et al., 2019), and therefore display an elevated spatial variability, extrapolating methane flux

results is associated with large uncertainties.

6.    Wintertime Methane fluxes (*Winter-CH₄*): this set is the intersect of the wintertime and methane fluxes set.

The core question we aim to answer for each of these subsets of sites is how well the existing network is capable of capturing spatio-temporal variability in environmental conditions, and therefore also in surface-atmosphere fluxes across the pan-Arctic domain.

**2.5 Upgrades to observational network**

One closely related task to evaluating current network representativeness is to identify the optimal locations for a coordinated network expansion, in case our analysis reveals substantial gaps in network coverage. Since testing each cell and each combination of a number of expansion locations would come at excessive computational costs, we developed the following approach for this purpose.

We first restricted new site locations to places with existing infrastructure, mainly villages and weathers stations. The reasoning for this was that setting up and servicing an eddy covariance site — especially when aiming at staying operational during wintertime — requires some level of infrastructure, and ideally staff that lives nearby. Thus we identified the locations of populated places within the Arctic as described in the natural earth populated places dataset (https://www.naturalearthdata.com/downloads/10m-cultural-vectors/10m-populated-places/). This first shortlist of potential

new sites was further reduced by excluding all villages in ecoregions that already contained an EC site in the *All* subset. This included some of the most densely populated Arctic regions, thus significantly reducing the number of potential new sites to just 109.

For the *Winter*, *Methane* and *Winter-Methane* scenarios, we opted for a different subset of candidate sites. As upgrading existing sites is far more cost efficient than establishing a new one, instead of using new locations we first focused on existing sites that lack either wintertime- or methane measurements, or both. A final stepwise analysis also included both existing and new candidate sites for these scenarios.

For each candidate location, we created an individual representativeness map that quantifies how similar each area around the Arctic is to those environmental conditions at the given site. To evaluate how the addition of each site, or combinations thereof, influences the overall representativeness of the observation network, one or several of these maps were subsequently combined with the existing representativeness maps of the different scenarios outlined above. Since the influence of multiple towers on a single pixel is not additive in our approach, but instead only the single best score will be retained, the final representativeness score on a pixel-by-pixel basis is simply the minimum value across all individual maps that are being combined. The overall impact of new sites being added was finally evaluated by comparing median representativeness scores across the Arctic region between original and extended network versions.

We tested three methods to quantify the impact of adding individual new sites, or combinations thereof, on the overall network representativeness score. Ranking these results allowed us to optimize the network based on these maps, i.e. identify those new sites that best complement the existing coverage.

1. *Exact*: this method tests all possible combinations of adding a set of k new sites to existing observational networks. It thus guarantees that, for each k value, the combination of new sites can be identified that optimally enhances overall network representativeness. It is highly computationally expensive though: for example, given a pool of 109 candidate sites, adding *k*=3 new sites implies that there are already 209934 potential combinations that need to be tested. Since this follows a factorial growth until k equals the size of half the dataset, the method is thus only applicable for a small number of additional sites.

2. *Stepwise*: instead of comparing all possible combinations when adding multiple sites, this approach sequentially identifies a single best site that can be added to an existing network. Starting with an existing network, all candidate sites are tested individually, and the one site is selected that results in the best improvement to the network representativeness. This site is then added to the existing network, and accordingly excluded from the list of candidate sites. In the next step, the approach searches among the remaining candidate sites for the next best addition, adds it to the existing network, and so on. This iteration continues until all candidate sites have been added in their order of relevance. While this simplified approach cannot guarantee that the combination of k sequentially added sites is indeed the best combination of k sites to be added to the existing network, it significantly reduces computational expenses, and therefore facilitates also the identification of subsets of sites also for large k values. For example, selecting k=3 new sites from our pool of 109 candidate sites this way just requires testing a total of 324 combinations, which is several orders of magnitude lower compared to the exact method.

3. Stepwise ecoregion exclusion (*stepwise-ee*): this method is identical to the *stepwise* method described above, only instead of removing just the single selected sites from the list of candidate sites, here we remove all sites from the same ecoregion as the selected site.

Owing to the excessive computational costs, the application of the *Exact* optimization method had to be limited to a low number of additional sites. Based on this method, we identified the best subsets of sites to be added, and the corresponding improvement in network coverage, for 1-3 new sites for the *All* scenario, and for 1-6 new sites for the remaining 3 scenarios. We therefore resorted to using the *Exact* method as a reference to evaluate the performance of the computationally more efficient, but only approximate *Stepwise* method, and found that both approaches yield corresponding results within the overlapping ranges. All further optimization results are therefore based on the *stepwise* results, since it allows evaluation of a larger subset of new sites.

To evaluate the efficiency of these guided approaches to upgrade existing observation networks with new sites, as a control we compared the results based on the approaches above with network upgrades using random selection of new sites. In this context, for each subset of new sites to be added to the network or to be upgraded, a total of 100 unique combinations of these candidate site sets were drawn, and the median of the observed increase to the network representativeness score was taken as the final result. Cases with a low number of new or upgraded sites, i.e. where the number of possible combinations was smaller than 1000, were excluded to warrant the randomness of sample drawing. Instead here for low values of k we used the median of all combinations as computed by the *exact* method since a sufficiently large sample of random tests approaches this value. The guided approach should see large gains in initial network development, as the most optimal sites are chosen first. Consequently, with the best locations already been selected, later additions will have a reduced impact on the network representativeness. Using a random site selection method, we expect initial improvements to be lower, but at the same time the decline in improvement per additional site will be less since later additions might still contain high impact locations. While normally sites are not strictly selected at random, they are typically not chosen with the entire network in mind, and some opportunism exists as far as accessibility, funding, and existing infrastructure.

**Section 3: Results**

**3.1 Assessment of flux site infrastructure**

Through merging information from existing databases, and adding details from the online survey among site PIs described above, we identified 120 EC sites situated within the domain north of 60 degrees latitude. 83 of these sites (69%) were listed as active at the start of 2019, while the remaining 38 sites had been either permanently or temporarily discontinued at that time. The distribution of these sites across the study domain is uneven, with the majority located in Europe and Alaska (61% of all active sites), i.e. regions that only account for about 12% of the total surface area. This imbalanced distribution of sites

315 (Fig. 1) leaves large regions of the Arctic with comparatively sparse network coverage, particularly regarding Central and Eastern Siberia, and Eastern Canada.

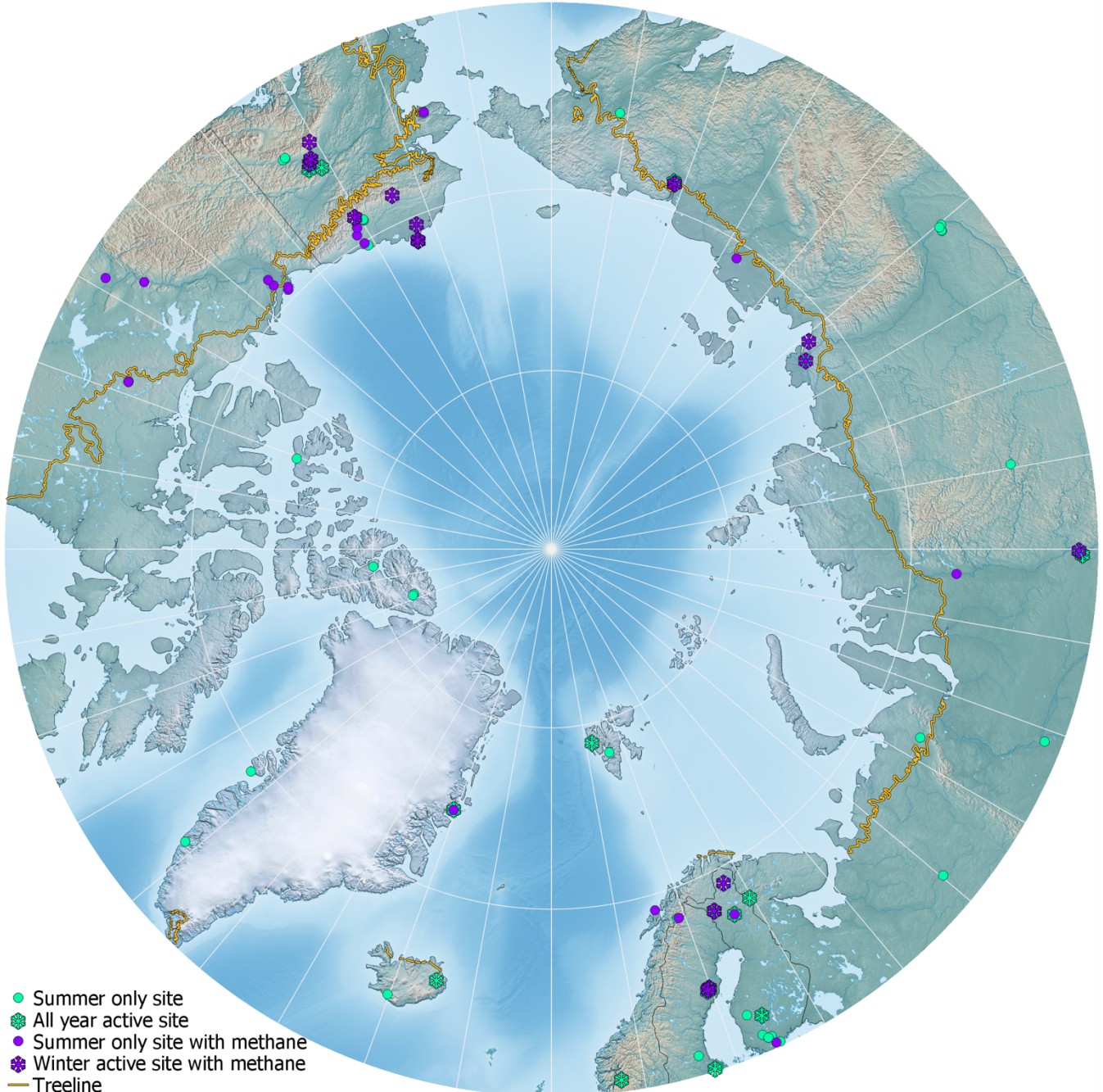

**Figure 1: Overview map of EC sites in our consolidated Arctic database. Green symbols indicate sites with CO$_2$ fluxes only, whereas purple indicates CO$_2$ and CH$_4$ flux measurements. Snowflakes show sites with reported wintertime measurements. The**

 **yellow line indicates the Arctic treeline (https://www.geobotany.uaf.edu/cavm/data/, status 28-02-2020), background Natural Earth 2.**

The number of sites within the Arctic EC network has steadily grown since the establishment of the first sites in Alaska in 1993: Utqiagvik (formerly Barrow), Happy Valley and Upad. Figure 2 indicates that the installation of new sites gained momentum in the late 1990s, and the network steadily grew until reaching its current level of slightly over 80 active sites around 2011. Since that time, the size of the network has remained more or less stable, i.e. newly established sites largely balanced site shutdowns. Owing to the harsh Arctic climate conditions, wintertime site activity is clearly lagging behind summertime data coverage. Of the 33 sites that report year-round activity, 25 sites are still in operation. Accordingly, year-round activity, i.e. sites including cold season data coverage, is currently at about the same level as the summertime measurements were 15 years ago. Moreover, 81% of these wintertime measurements took place in Europe and Alaska, leaving most parts of Canada and Russia with very low data coverage outside the growing seasons.

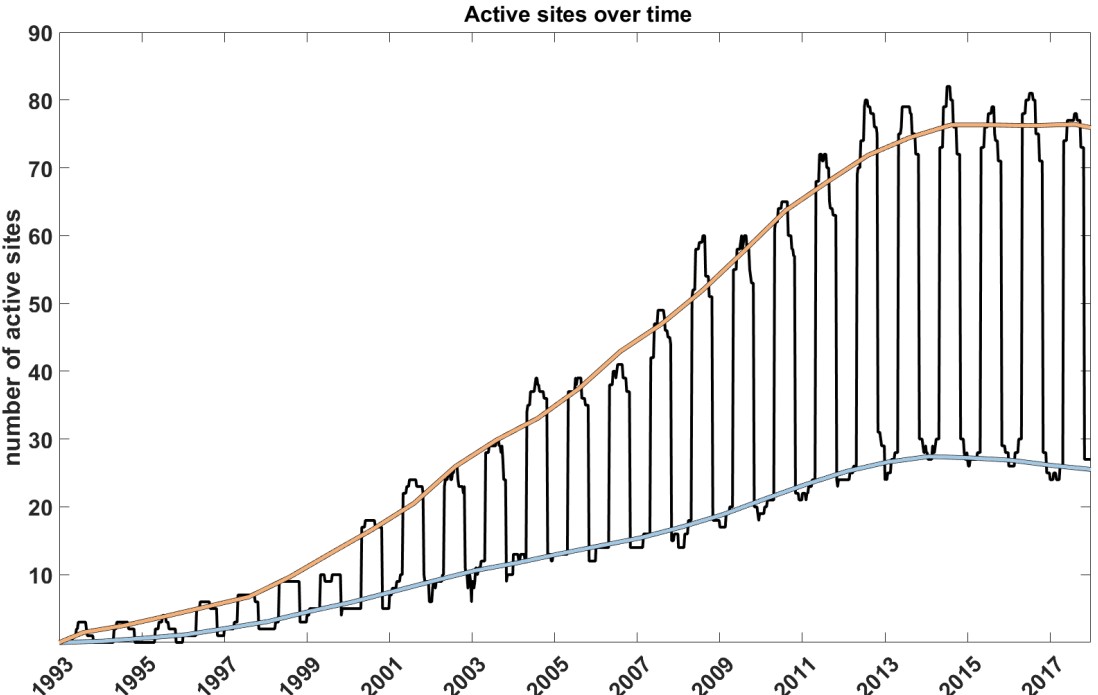

**Figure 2: Development of eddy-covariance network data coverage at monthly time steps. The fluctuating black line gives the total number of active sites per month, orange and light blue lines indicating the long-term development of data coverage during summer and winter, respectively. For sites from our dataset where activity was only specified per year, summertime-only data coverage was assumed.**

Regarding the length of the time series covered by the eddy flux sites, there is a pronounced variability across the network. The longest running site (US-Brw: Utqiagvik, formerly Barrow) has been active for 28 years at the time of writing. Due to

the substantial extension of the network in the 2000s, today the median activity among all sites is 8 years. The steady increase in the length of the data records over time implies that a growing number of sites is suitable to detect trends in flux

rates that can be linked to ongoing climate change at site level across the Arctic.

**Table 1: Overview of activity of EC sites in the study domain (>= 60ºN) by 2019. For sites that did not report data availability on a monthly basis, we assumed no activity during wintertime.**

| Subset | Active | Inactive |
|---|---|---|
| *All* | 83 | 37 |
| *5 year* | 73 | 16 |
| *Winter* | 33 | 8 |
| *$CH_4$* | 32 | 14 |
| *Winter - $CH_4$* | 16 | 3 |

Regarding the measurement of non-$CO_2$ fluxes, only for methane a considerable number of observation sites could be identified that provides longer-term flux data coverage. Even though the methane network has been growing steadily over the past years owing to the availability of a new generation of gas analyzers, the number of sites at which $CH_4$ fluxes are monitored is lagging far behind the $CO_2$ data coverage: 2019, only 32 active sites were identified, 14 (30 %) of which were inactive. This is similar to the wintertime data coverage: even though methane flux data coverage has been improving over

recent years, there are still large gaps in the network, and data coverage is at about the level the $CO_2$ summertime data featured in the early 2000s. For other non-$CO_2$ gases, such as $N_2O$, no observational infrastructure could be identified in the context of our data survey.

Data availability is a crucial factor when it comes to the usefulness of the eddy-covariance observations for community-wide research efforts in the context of climate change. PI responses to our survey indicated that the majority of the eddy-

covariance datasets is currently available to interested users: 18% of the datasets were reported as open access, and a further 44% will be made available on request. Thirty six percent of the datasets comprising our database are still being processed and/or reviewed by the site PIs, but will be made available in the future. Only a small fraction (2%) is not intended to or can no longer be shared publicly.

**3.2 Representativeness assessment**

Our analysis of the representativeness of the Arctic EC tower network reveals pronounced regional gradients. The choice of the subset of towers (Fig.3) clearly shows the difference in representativeness. At the same time, also the two different

quality standards (Fig. 3 and 4) show stark contrasts when differentiating the domain into represented and upscalable areas. Linked to their dense coverage with continuously operated sites, across scenarios the northern European countries Finland and Sweden as well as the North Slope region of Alaska stick out with the highest data coverage. At the other end of the coverage spectrum, the representativeness analysis of the Arctic EC site network shows large areas of Siberia and Canada as poorly represented (Fig. 3 and 4), even when it comes to summertime data of $CO_2$ fluxes.

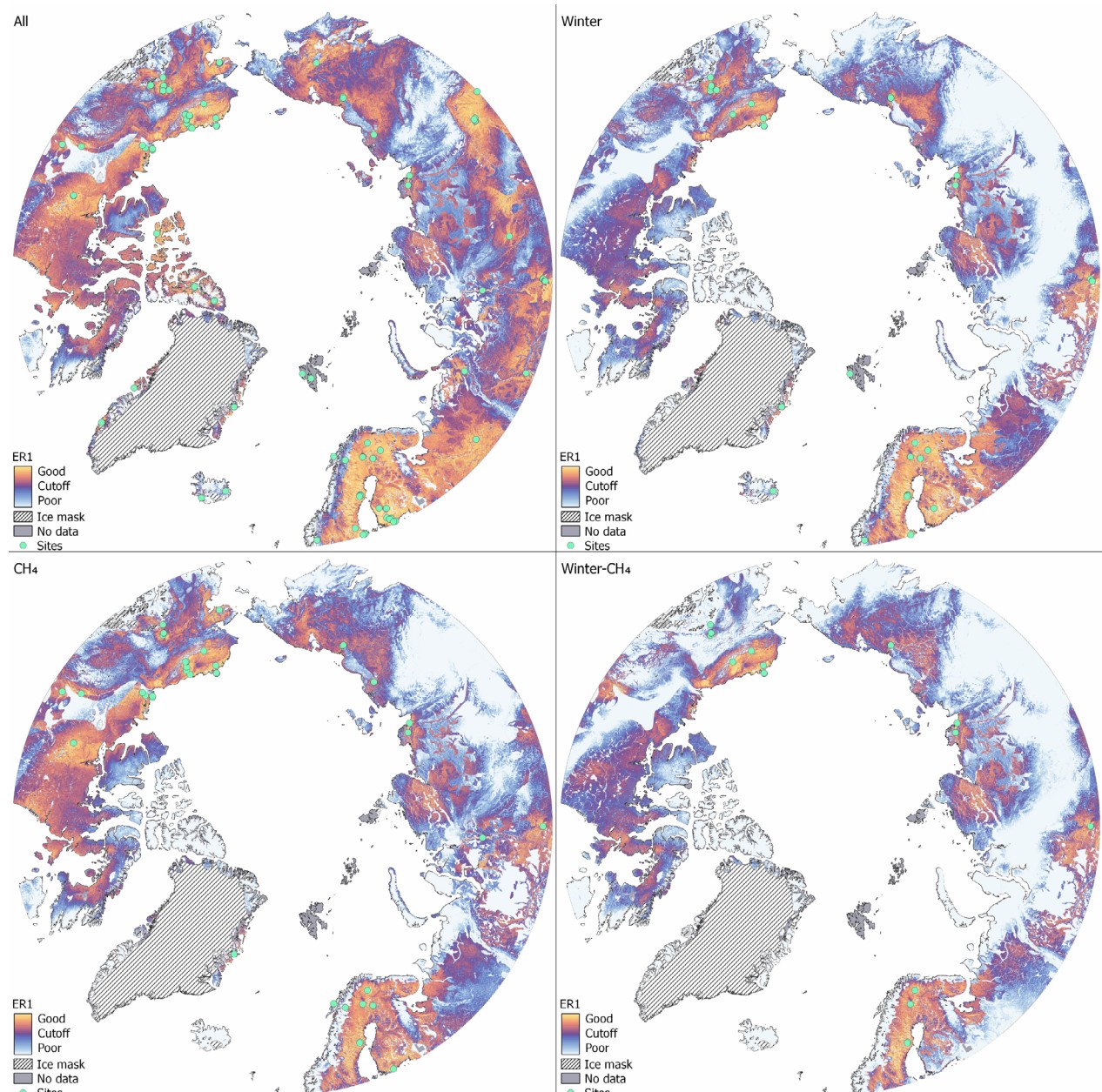

**Figure 3: Representativeness of *All (top, left), Winter (top, right)*, *CH₄* (bottom, left) and *Winter-CH₄* (bottom, right) fluxes subsets.** The representativeness of *Active* and *5-year* subsets have a similar pattern though reduced values as the *All* scenario and can be found in the Appendix C. The center value of the color spectrum equals the ER1 cutoff (i.e. the 75[th] percentile of representativeness values of ecoregions with at least one site), thus warm/yellow colors match represented regions whereas cold/blue colors do not meet this criterion.



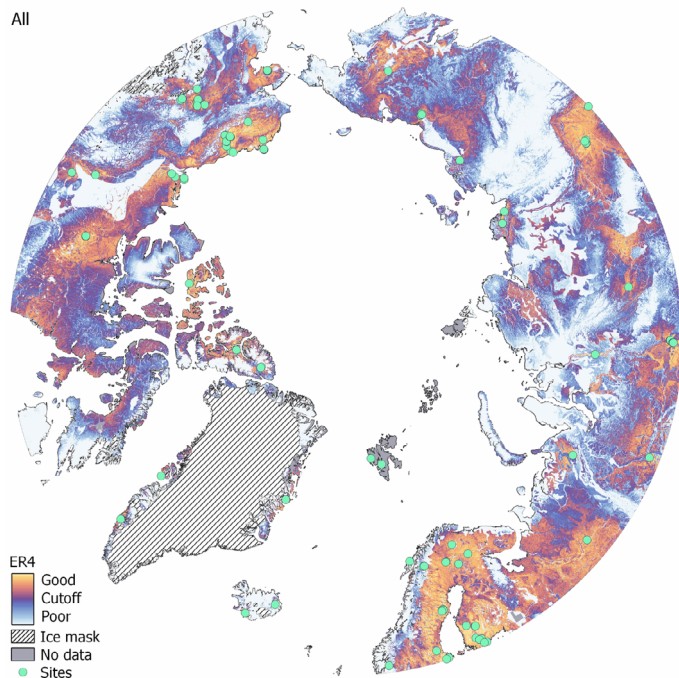

All

ER4
Good
Cutoff
Poor
Ice mask
No data
Sites

**Figure 4: Representativeness of the *All* subset of sites using the ER4 cutoff as a center value of the color spectrum (i.e. the 75[th] percentile of representativeness values of ecoregions with at least four sites). As in Fig. 3 above, warm/yellow colors indicate regions with a score below this cutoff, i.e. well represented regions, whereas cold/blue colors do not meet this criterion. The**
**representativeness values underlying this map are identical to those used in Fig. 3 above, but due to the stricter ER4 quality criteria, the size of those domains that fall within this cutoff is lower. However this region can realistically be upscaled from the existing network .**

The location of coverage gaps in our representativeness maps can to a large extent be explained by ecosystem characteristics.
The majority of EC towers located >=60 degrees north that were included in our study are either located in lower lying tundra landscapes and wetlands, or in forests of the taiga sections included in our domain. Higher elevations, particularly mountain ranges, generally show a low EC flux data coverage across the Arctic. A comparison of our representativeness scores with Arctic DEM (Porter et al., 2018) elevation data as a proxy for mountain ranges resulted in a positive correlation (r =.26, p<0.001). Accordingly, the majority of the larger gaps indicated by our maps are characterized by higher elevation.
We see large differences between the tested subsets. Encompassing the full 120 sites within the database, the *All* network produces the largest fraction of represented areas, while the *Active* current network status and the *5-year* networks, both based on a considerably smaller number of towers, differ only slightly in overall coverage and regional patterns. Linked to the lower number of applicable tower sites, *wintertime* activity and $CH_4$ measurements show a pronounced reduction in network coverage in comparison. This emphasizes the outstanding character of the previously mentioned highly
instrumented regions even further: A high regional representativeness for methane fluxes is mostly limited to the Alaska

North Slope, the Fairbanks region, Sweden, and Finland. Outside these regions, representative data coverage is only sporadic. Regarding wintertime measurements, a similar picture emerges as described for methane fluxes, but here some extra sites in Canada enhance network coverage in this domain.

The ER1 and ER4 metrics described in Section 2 can be used to quantify the fraction of the study domain that falls within
the specified cutoff values. Based on the ER1 metric, about half of the Arctic terrestrial area can be considered as being represented by at least one tower for the *All*, *Active* and *5-year* networks (Table 2). The fraction of the domain that is represented drops to about one third for methane measurements, and is even lower for the wintertime observation network (26%). Finally, wintertime methane measurements only cover one fifth of the Arctic. Based on the ER4 case aiming at minimal required upscaling standards, all these values get further reduced. In this case, the *All*, *Active* and *5-year* site
networks can only reliably be upscaling to about one third of the Arctic domain, whereas *Winter* and *CH4* measurements can represent only 13% and 19% of the domain respectively. With only 9% coverage, wintertime methane is largely limited to the Alaska North Slope and Sweden. With this more restrictive metric, the direct local influence of individual towers becomes more apparent.

A comparison of the CAVM map with the k-means clustered ecoregion maps shows 52% percent of the grids are identical,
whereas when we look for similar vegetation (e.g. Cryptogam, herb barren with Cryptogam, barren complex) we find them to be in accordance in 66% of the grids. Figure B1 shows a cluster-based visualization of the comparison.

**Table 2: Representativeness, percentage difference, ER1 and ER4 for the 6 subsets.** *Representativeness* **indicates the median representativeness values of the entire domain; the closer to zero the better the representativeness;** *% diff* **the difference in**
**representativeness compared to the** *All* **scenario, with a larger difference indicating a lower representativeness compared to the entire network in its** *All* **subset.** *ER1* **and** *ER4* **represent the fractions of the domain that fall within their respective cutoff values.**

| | Representativeness | % diff | ER1 | ER4 |
|---|---|---|---|---|
| *All* | 0.0081 | 0% | 0.55 | 0.35 |
| *5 year* | 0.0089 | 9% | 0.50 | 0.32 |
| *Active* | 0.0094 | 17% | 0.46 | 0.28 |
| *Winter* | 0.0141 | 73% | 0.26 | 0.13 |
| *CH4* | 0.0120 | 48% | 0.34 | 0.19 |
| *Winter-CH4* | 0.0159 | 95% | 0.21 | 0.10 |

### 3.3 Upgrades to observational network

The targeted selection of new site locations to sequentially fill the biggest gaps in the existing network, as executed by these
stepwise optimization approaches, yields clear improvements in overall network coverage (Fig. 5) compared to the conditions before the optimization (i.e. the 'current' network as shown in Figures 3, 4, and Table 2). For example, when adding 15 new sites to the *All* network (an increase in number of sites by 12%), with the guided optimization we could increase the fraction of the domain which falls within the ER1 cutoff rise from 55% to 69%, corresponding to a relative increase of 25%. Since the other three scenarios start at an overall lower coverage level for the existing network, increases in
both the fraction of pixels that meet the ER1 criteria as well as their percentage change are larger (Table 3). Particularly for the *Winter* and *Winter-CH$_4$* networks, the area could approximately be doubled, or more than doubled, respectively. Gains in overall network coverage are biggest for the first sites added, then asymptotically level off (Fig. 6). For the *All* network, this pattern is rather subtle, while for the other three scenarios, the flattening of the curve is clearly visible after about 10 sites have been added. This can be attributed to the fact that the pool of candidate sites is significantly higher for the *All* network,
with 109 potential new locations whereas *Winter, CH$_4$ and Winter-CH$_4$* have 30, 25, and 38 potential upgrade sites respectively. The fact that the *All* network is already better represented is another difference that can explain the more gradual and smaller relative improvements. Network coverage fractions are different for the ER4 metric, while in relative terms the gains in numbers when adding new sites are comparable to the ER1 results. It is of interest to note that as few as 5 site to upgrade the network can double the size of the region of the *Winter-CH$_4$* network that meets the ER4 criteria. Between
15 to 25 new site additions to the *All* subset would be required to have a high (ER4) statistical confidence for upscaling that covers 50% of the Arctic terrestrial domain.

**Table 3: Fraction of the domain that meets the ER1 and ER4 criteria when sequentially adding up to 25 new sites to the networks, broken up into four network scenarios.**

|      | ER1  |        |        |            | ER4  |        |        |            |
|------|------|--------|--------|------------|------|--------|--------|------------|
|      | *All* | *Winter* | *CH$_4$* | *Winter-CH$_4$* | *All* | *Winter* | *CH$_4$* | *Winter-CH$_4$* |
| Base | 0.55 | 0.26   | 0.34   | 0.21       | 0.35 | 0.13   | 0.19   | 0.10       |
| 5    | 0.63 | 0.41   | 0.47   | 0.36       | 0.42 | 0.23   | 0.28   | 0.20       |
| 15   | 0.69 | 0.5    | 0.52   | 0.46       | 0.48 | 0.30   | 0.32   | 0.27       |
| 25   | 0.72 | 0.52   | 0.53   | 0.48       | 0.52 | 0.32   | 0.33   | 0.29       |


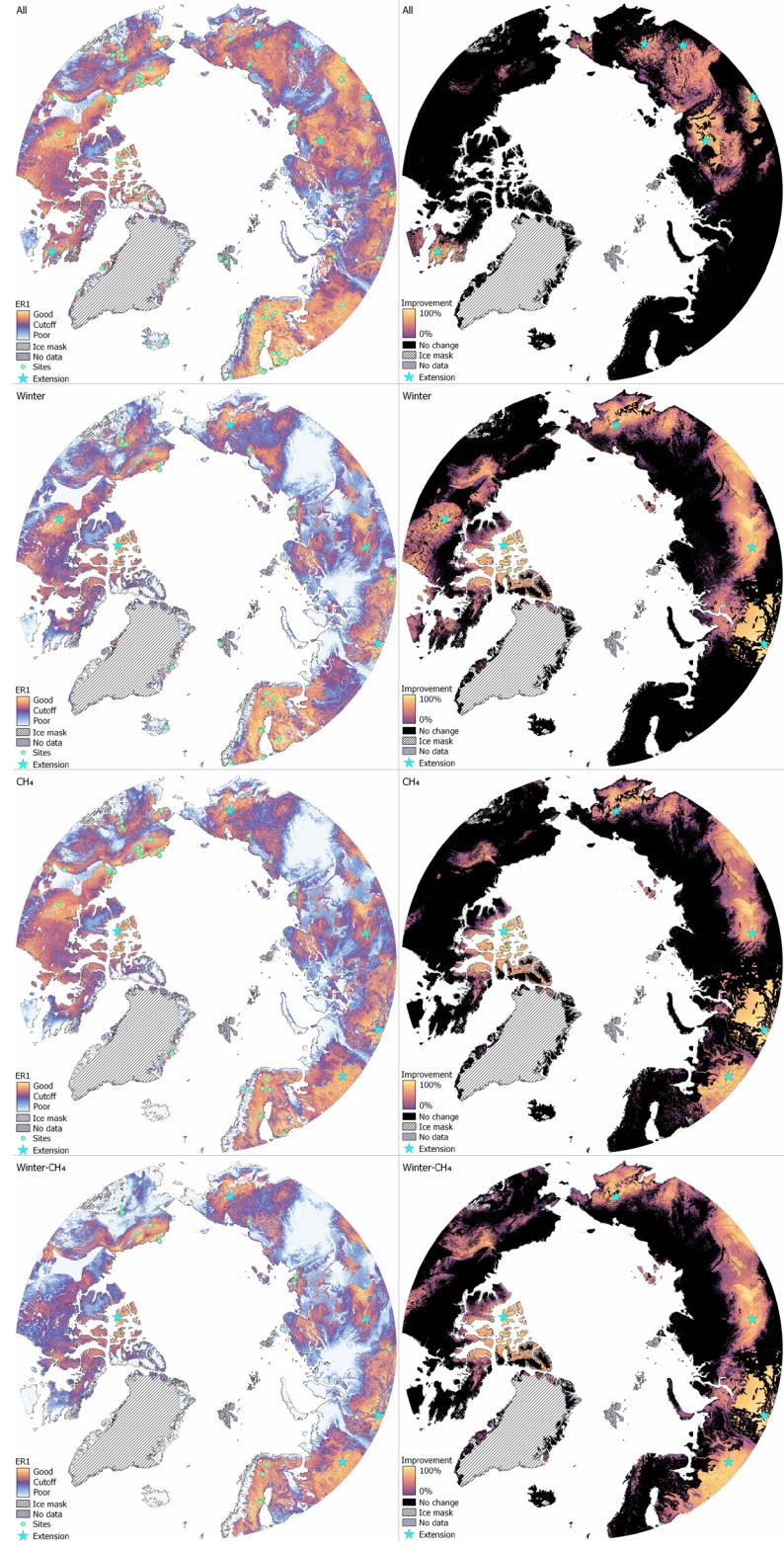

**Figure 5: Network improvement compared to original coverage after adding 5 sites with guided optimization. (Left) As in Figure 3, the center value of the color spectrum equals the ER1 cutoff (i.e. the 75th percentile of representativeness values of ecoregions with at least one site), thus warm/yellow colors match represented regions whereas cold/blue colors do not meet this criteria. Green dots represent existing sites. Stars represent the location of selected upgrade or extension location. (Right) Relative improvement in network coverage, compared to pre-optimized conditions. Here, orange colors indicate a large relative improvement whereas purple indicates minor improvement. Black areas experienced no change.**

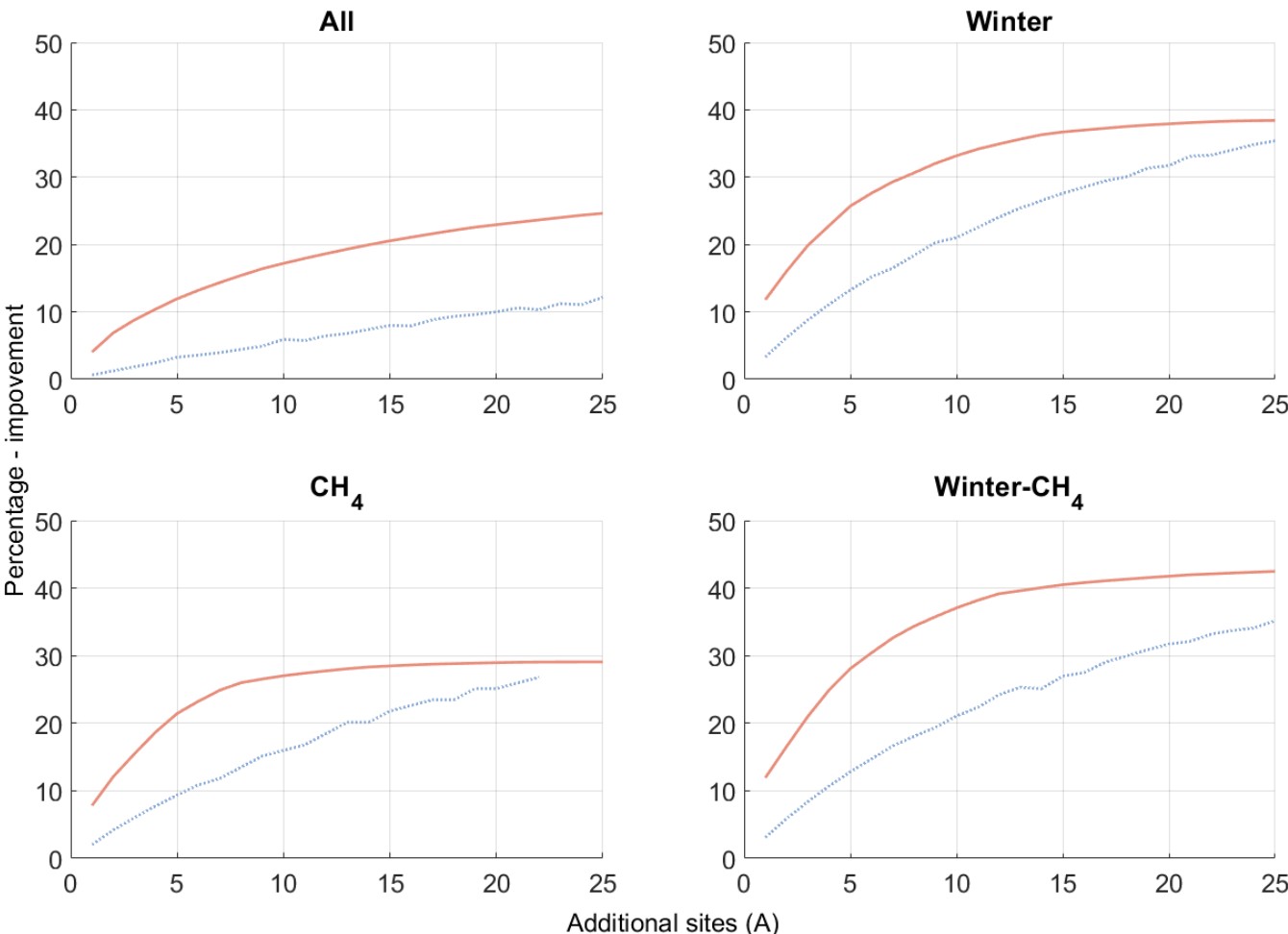

**Figure 6: Percentage improvement compared to pre-optimized for the first 25 additions. As opposed to table 3, these panels show improvement in the representativeness value, and not the ER1 or ER4 Metrics. Orange lines indicate the improvement with the selective stepwise optimization method, blue dotted lines represent the alternative approach of improving the network by selecting sites in random order.**

While using the presented optimization method, gains in the representativeness of the domain are characterized by an initial steep increase followed by a gradual levelling off, these gains follow a near-linear trajectory when randomly selecting additional sites (Fig. 6). This is clearly reflected by the mean network improvement that is imposed by the first site added: for example, in case of the *All* subset of sites, identifying the best site to be added improves the entire network coverage by 4.0%, whereas any random site would on average only lead to an improvement of 0.6%, a difference by a factor of 6.7. This advance gradually decreases, where the number of sites to be added is close to the number of potential sites the overall improvement between a targeted optimization and a random site selection are close again, since at such a point there no real choice to be made. The greatest cumulative difference between the optimized and random assignment of sites is found at around 10 sites to be added, but even for high numbers of new sites the optimization method will always perform better than the random method.

Comparing the *stepwise* method against the *stepwise-ee* method that excludes candidate sites within ecoregions that have already been filled with a new site, we found the former to produce slightly better results. For the *Winter* and *Winter-CH$_4$* scenarios, there are no differences between these methods for the first 10 sites that were added. For the *CH$_4$* scenario, with the *stepwise* method only a single site was chosen that was excluded in the *stepwise-ee* approach, resulting in a 0.1% difference in the fraction of represented pixels. Substantial differences were only found for the *Active* scenario, where four sites were selected by *stepwise* that were excluded by *stepwise-ee*, however, even in this case the net difference in network coverage was just about 1%.

For *Winter*, *CH$_4$*, and *Winter-CH$_4$* scenarios, besides restricting network extensions to existing tower locations for reasons of cost efficiency (*Upgrade* approach), we also conducted a network optimization using all available candidate locations (*New+Upgrade* approach). In this context, we found that the *New+Upgrade* approach yields higher gains in the fraction of represented pixels by an average of 0.7% per added site compared to the *Upgrade* scenario.

To evaluate the robustness of this optimization, and investigate whether or not small changes in experiment setup may lead to vastly different results, we used the output of the *Exact* optimization method in additional experiments. This method not only produces a single combination of sites that offers the best network improvement, but investigates all possible combinations and their impact on the network. For the actual test, we compared the ecoregions that included the best subset of sites, with those ecoregions selected in the top 100 subsets. We found that, on average, 74.6% of the ecoregions included in the top-100 list match the regions selected for the optimum case. Accordingly, even though different subsets of sites were selected, the regions targeted for extension remained largely the same, as did the quantitative gains in network coverage.

## Section 4: Discussion

### 4.1 Assessment of flux site infrastructure

This study documented the past and current status of the Arctic eddy-covariance site infrastructure, assessed current gaps in network coverage, and developed strategies on how to best fill them. Analyses were based on metadata on the pan-Arctic EC network summarized in an online mapping tool, demonstrating the expansion of the network since its inception in 1993 to 120 individual tower locations. We show here that even though the network has expanded substantially over the past decades, there are still large coverage gaps. These gaps concern not just spatial representation of the heterogeneous Arctic landscape, but also the monitoring of key parameters such as methane, or temporal aspects such as wintertime and zero curtain fluxes.

While great care has been taken in collecting metadata for our database of Arctic flux sites, this database by its very nature is a work in progress. Accordingly, the current state of the online database will deviate slightly from the version used in this paper, since we continuously work in data updates provided by site PIs, and also encourage PIs of new sites to contact us in the future. For reference, a version of the database reflecting the state that was used to produce results summarized in this study has been retained. Since we rely on PI feedback to ensure correctness of the collected information, occasional gaps and outdated data in the database are possible.

Since we did not receive PI feedback on our database survey for some sites, we do not have information on site activity in monthly (or seasonal) time steps for the entire Arctic network. For sites where this information is missing, we assume summertime activity only, i.e. no non-growing season flux data is available. This assumption is based on an assumed workflow where in spring, once sites become accessible again, the equipment is serviced and activated for operation during the growing season, and then is kept running into autumn and winter until instrument failures and/or loss of energy supply terminate data acquisition. As a consequence, the site lists used for the non-growing season represent a conservative picture of the year-round network coverage, i.e. we only consider those sites for which wintertime activity was confirmed. We anticipate refining this assessment with additional PI responses into our database survey. In Figure 2, the network growth seems to level off around 2012. However, it is possible that part of this slowdown in network growth can be attributed to delays in updating sites and studies in the online depositories. Since it is not uncommon to restrict data access until first results have been published by the data owners, future data availability for the most recent years may in fact be higher than reflected by our database.

We would like to highlight that while the main focus of our analysis was on the spatial pattern of measurements, the temporal distribution of measurements is an important aspect as well, and not all temporal effects could be captured by the method applied for this study. On a short temporal scale, data gaps are a problem that needs to be resolved when calculating annual budgets. A typical tower in temperate climate zones has a data coverage of 65% (Falge et al., 2001), and considering the typical winter time shutdown and more extreme weather, this value will be lower in the Arctic. And while there are gap

filling methods, the errors of these methods increases with gap size (Falge et al., 2001; Moffat et al., 2007). Furthermore, it should be noted that long time series are exceptionally valuable for studying ecosystem feedbacks with climate, as explained e.g. by Baldocchi (2020), and especially interannual variability and long term ecological trends are impossible to detect without long term observations.

## 4.2 Representativeness assessment

Our evaluation of the representativeness of different subsets of EC stations was based on a pixel-by-pixel comparison of bioclimatic and edaphic conditions between tower locations covered by the network and the Arctic study domain. Evaluating all available sites, we obtained sufficient coverage by the ER4 metric for Finland, Sweden, Western Russia, Alaska and parts of Canada, while large regions of Canada and Siberia were poorly represented. This matches our earlier observations evaluating the general distribution of site locations. Besides this regional imbalance, across the Arctic large coverage gaps were associated in mountainous regions.

Focusing on the ER1 metric — which shows a representation similar to an ecoregion with at least one tower — only about half of the Arctic (excluding glaciers) can be considered represented by the EC tower network, as far as $CO_2$ fluxes are concerned (Table 2, *All, 5 year, Active*). Limiting site selection to subsets *Wintertime* and *CH₄*, represented regions were substantially reduced to about a quarter of the Arctic (Table 2, *Wintertime, CH₄, Wintertime-CH₄*), and largely focused on Finland, Sweden, and parts of Alaska. A focus on the ER4 metric indicates that only one third of the Arctic can be represented with a high statistical power (Table 2, *All, 5 year, Active*), and if we consider the wintertime networks as the configurations with the only reliable year round carbon budget, this value drops to about a tenth of the Arctic domain. This constitutes an important gap in data coverage, since while wintertime fluxes in the Arctic are substantially lower than those during summer time, they are still significant for Arctic carbon budgets (Zimov et al., 1996; Wille et al., 2008; Euskirchen et al., 2012; Marushchak et al., 2013; Lüers et al., 2014; Oechel et al., 2014; Zona et al., 2016; Natali et al., 2019).

Comparing our representativeness assessment with similar works shows a good match with results presented by Virkkala et al. (2019), who also identified best data coverage for Fennoscandia and Alaska, while the overall patchy coverage in Siberia mainly focused on individual, densely instrumented research stations. Also a global network evaluation (Jung et al., 2020), based on a so-called extrapolation index as an indicator of expected error, shows a similar pattern, with Arctic errors generally at a high level, compared to the global average, but Canada and Siberia showing exceptionally high extrapolation uncertainties within the Arctic. And while only having a small overlap with our domain we see a similar under-representation of Norwegian mountain regions as shown in Sulkava et al. (2011).

Our evaluation of network representation provides valuable information for flux synthesis, upscaling or data assimilation activities. When upscaling fluxes, our maps can be utilized as a measure representing the extrapolation uncertainty from observation sites to the larger domain. For the current network, these maps make it obvious that EC data can reliably be upscaled within Fennoscandia and Alaska, at least when average bioclimatic and edaphic conditions are considered, while

within other domains special care is required regarding site selection, and weighing their inputs, in order to avoid systematic bias. In addition, when using upscaled flux fields as prior input in atmospheric inverse modelling studies to constrain greenhouse gases, the representativeness maps can be utilized to constrain a priori error maps estimates.

The network representativeness analysis presented here is powerful in showing the patterns associated with the networks coverage because the analysis is based on key climatic, soil, and topographic variables, and especially takes into account Arctic-specific controls such as permafrost extent. However, the assessment of specific fluxes provided by the eddy covariance tower network based on these data layers must largely remain qualitative, since no clear quantitative linkage between the bioclimatic controls and the fluxes for $CO_2$ and $CH_4$ can be considered. In other words, assigning equal weights to all 18 data layers allows us to assess a general level of similarity between pixels within the domain, but does not necessarily reflect how strongly potential differences will influence greenhouse gas flux rates (see e.g. Tramontana et al., 2020) The good fit between the CAVM map and the ecoregion map further strengthens the choice of these 18 bioclimatic variables, as similar patterns and vegetation distributions are found with largely different methods. Differences between these maps are to be expected since the CAVM map has been produced using a far more extensive method (Raynolds et al., 2019). Furthermore, even within pixels there can be large variation that both methods cannot capture, but that influence the final classification of the pixel.

### 4.3 Role of small-scale variability

This evaluation functions on the premise that an EC site represents a specific type of ecosystem, as is generally the practice when working with EC data, and that the obtained flux data can be upscaled to the same ecosystem type within a larger region (Belshe et al., 2013; Olefeldt et al., 2013; Hill et al., 2017). However, the study by Hill indicates that even seemingly homogeneous ecosystems are subject to flux variability linked to minor differences in site characteristics such as e.g. exposure elements, nutrient availability or water storage capacity. Accordingly, to represent an ecosystem with more certainty, often more than one EC site needs to be installed. This finding emphasizes the value of the high-density networks of towers installed in Northern Europe and Alaska, where multiple towers capturing fluxes within the same type of ecosystem contribute to reducing uncertainties related to upscaling, and therefore improving data quality of the regional flux budgets.

Our representativeness evaluation is based on the assumption that each tower perfectly represents those conditions that are given for its specific pixel within the gridded maps used to evaluate Arctic landscape variability. In reality, however, many towers will be subject to variability in ecosystem characteristics within the field of view of the flux instruments, and data may thus only partially represent those averaged conditions given for the larger grid cell. The influence of sub-grid variability might be particularly important for methane fluxes, which are very dependent on local topography (Peltola et al., 2019) and especially water levels. To address the uncertainty linked to subgrid-scale variability, ideally for each tower a footprint analysis (Göckede et al., 2008) would be performed that allows quantification of the representativeness of the tower

location for the ecosystem characteristics listed in the gridded maps. As the complexity of a landscape increases, so would the importance of an analysis like this. Boreal forest can show a surprising amount of heterogeneity (Ylläsjärvi and Kuuluvainen, 2009), and the polygonal nature of some Arctic tundra landscapes (Virtanen and Ek, 2014) makes it exceptionally difficult to arrive at one value for the entire region with just one tower. Mobile towers that can be easily relocated to study heterogeneity in flux rates within a structured landscape (Sturtevant and Oechel, 2013), may be a solution to address this. As an alternative, the installation of a cluster of towers with low budget equipment as mentioned in Hill et al. (2017) would be another option to address spatial variability. We are aware of this problem, but lack the database to quantify it at this time. For some locations in this study, there are multiple towers within the same pixel, therefore we can capture the effect of small-scale variability. Most towers, however, are the sole measurements in their pixel. Over the coming decades, gridded products based on satellite observations are expected to increase in availability, and also their spatial resolution will improve. This can grant opportunities in the future to look at current sub-pixel heterogeneity, or simply assess variability at such a small scale that subpixel heterogeneity is no longer a serious concern.

Finally, a limitation of using gridded maps is that a majority filter may discriminate against minor landscape elements, which rarely are widespread enough to cover an entire grid cell. If such elements are 'lost' in the gridded maps, we would overlook an aspect of landscape variability. Therefore, in this study the representativeness of the methane scenario should be considered a best-case scenario.

**4.4 Upgrades to observational network**

Across different optimization methods tested herein, we could demonstrate that our site selection strategy targeting least represented regions within the study domain was clearly superior to unguided site selection regarding the improvement of overall network representativeness. Independent of the subset of network to be upgraded, the majority of the new towers were placed in Russia, with the remaining ones used to fill coverage gaps in Canada. For example, adding just 10 additional towers resulted in about 35% improvement for W*inter* flux coverage, and 30% improvement for $CH_4$ fluxes. Furthermore, our results demonstrated that upgrading existing sites to either measure new GHG species or remain active during wintertime led to similar enhancements in the specific subset network coverage than establishing new sites, at considerably lower costs. For the *All* scenario we opted to only consider ecoregions for expansion that did not have an EC site. This reduced the number of candidate locations from 348 to 109 sites, which also helped to reduce computational costs. However, multiple sites in a region can result in better representation scores, and accordingly we identified some cases where the *Stepwise* method recommends to add several sites within a single ecoregion as the optimum solution to improve the network coverage. However, comparing the *stepwise* with the *stepwise-ee* method demonstrates that differences are small. This indicates that our exclusion of candidate sites within regions that already contain an EC tower should only have had minor impacts on the performance of this network analysis, and can be justified given the gains in computational efficiency.

Concerning the $CH_4$ flux network, upgrading existing sites that already measure $CO_2$ fluxes is only marginally less effective than creating entirely new sites. Since the costs of upgrading an existing site with a methane analyzer is between 9 – 28% of the investment required for establishing a new site (ICOS ERIC, 2020), the savings from focusing on the more cost efficient upgrading strategy outweighs the gains in network coverage obtained from wider search options by far. Regarding the upgrade of an existing site for wintertime activity, there are less reliable numbers regarding the required investments. To keep a site running throughout the winter, extra power is required to heat or defrost instruments. At the same time, batteries are less reliable under cold conditions, and off-grid power generation can far less rely on the commonly used solar panels or wind turbines during the long and harsh Arctic winter. However, any new site that should stay active year-round will also need to cover such expenses. With only a 0.7% gain in representativeness through the higher degrees of freedom when also selecting new site locations instead of only upgrading existing sites, the savings linked to existing infrastructure and instrumentation provided by existing sites should outweigh the performance losses also for wintertime flux measurement networks.

**Section 5: Conclusion**

The Arctic is warming and changing rapidly, with implications both for global climate change trajectories and the livelihood of local communities. Large investments into adequate research infrastructure is required in the future to improve understanding of these Arctic changes across all relevant scales, and support the development of mitigation and adaptation measures. To efficiently use resources for an optimum upgrade of observational facilities, we need to advance our understanding on what our current measurements represent, and where gaps remain.

This study helps to guide efficient upgrades of the Arctic greenhouse gas monitoring facilities, showing that even though the Arctic EC network has grown considerably over the past decades, only half of the Arctic territory is represented by an EC tower at all, and this value drops to one third of the domain when we consider a statistically rigorous number of EC towers for upscaling. In particular, coverage within Siberia, Canada and mountainous regions is lacking. There are also large gaps when it comes to year-round data coverage, and non-$CO_2$ fluxes, with less than 20% of the Arctic terrestrial domain currently being covered by these measurements. While these numbers are associated with considerable uncertainties since we do not directly quantify how differences in ecosystem characteristics translate to fluxes, the applicability of this approach has been demonstrated by numerous previous extrapolation studies using similar underlying data as their explanatory variables. Accordingly, data-driven upscaling of EC databases to produce pan-Arctic greenhouse gas budgets, training datasets for biosphere process models or prior flux fields for atmospheric inverse modelling are still associated with large uncertainties, given the size of the regions currently underrepresented. We propose and test several methods for optimizing the EC network based on this representativeness assessment, and provide recommendations on network upgrades based on the best-performing and most practical option. Overall, as the most cost-efficient strategy for network improvements, we recommend upgrading selected existing locations with new instrumentation for methane measurements, since large coverage

gaps for this important greenhouse gas currently severely compromise our ability to comprehensively monitor carbon release from degrading permafrost within the extensive Arctic landscapes. Furthermore, keeping sites operational during the winter has been shown to be essential to understand annual carbon budgets within the Arctic, and also in this context winter-proofing strategically selected existing sites would provide the most efficient pathway towards better network coverage. A final step would be to extend the network further, especially in Siberia and Canada, and our method can help with selecting those locations that improve overall network coverage best.

This study, and associated datasets, has been designed to help the Arctic research community in planning future Arctic EC stations, and improve the quantification of uncertainties in the context of upscaling activities. Future studies could expand upon this study by selecting hard to sample regions, such as ecoregions without any current sampling and without villages or infrastructure, as a target for temporary towers or flight campaigns to empirically assess their (dis)similarity to already sampled ecoregions. Seasonal campaigns in mountainous regions could verify the assumption that fluxes in high latitude high elevation regions are low enough not to warrant the high investment and operation costs of permanent towers there. An assessment of heterogeneity in domains where replicative EC studies have been performed might provide guidance in better quantifying ecoregions sizes in data space for representativeness comparison.

**Table A1 Bioclimatic, edaphic and permafrost variables used for assessment of quantitative representativeness of Arctic EC network.**

| Variable Description | Units | Source |
| --- | --- | --- |
| Annual mean temperature for 1970-2000 | °C | WorldClim 2 doi: 10.1002/joc.5086 |
| Mean diurnal temperature range for 1970-2000 | °C | WorldClim 2 doi: 10.1002/joc.5086 |
| Isothermality for 1970-2000 | - | WorldClim 2 doi: 10.1002/joc.5086 |
| Temperature seasonality | °C | WorldClim 2 doi: 10.1002/joc.5086 |
| Mean Temperature of Warmest Quarter for 1970-2000 | °C | WorldClim 2 doi: 10.1002/joc.5086 |
| Mean Temperature of Coldest Quarter for 1970-2000 | °C | WorldClim 2 doi: 10.1002/joc.5086 |
| Annual Precipitation for 1970-2000 | mm | WorldClim 2 doi: 10.1002/joc.5086 |
| Precipitation Seasonality for 1970-2000 | mm | WorldClim 2 doi: 10.1002/joc.5086 |
| Precipitation of Wettest Quarter for 1970-2000 | mm | WorldClim 2 doi: 10.1002/joc.5086 |
| Precipitation of Driest Quarter for 1970-2000 | mm | WorldClim 2 doi: 10.1002/joc.5086 |
| Available Water Holding Capacity of Soil | mm | Saxon et. al. 2005 doi: 10.1111/j.1461-0248.2004.00694.x |
| Bulk Density of Soil | g/cm$^3$ | Saxon et. al. 2005 doi: 10.1111/j.1461-0248.2004.00694.x |
| Soil Carbon Density | g/cm$^3$ | Saxon et. al. 2005 doi: 10.1111/j.1461-0248.2004.00694.x |
| Total Nitrogen Density | g/cm$^3$ | Saxon et. al. 2005 doi: 10.1111/j.1461-0248.2004.00694.x |
| Compound Topographic Index | - | Saxon et. al. 2005 doi: 10.1111/j.1461-0248.2004.00694.x |
| Mean Annual Ground Temperature 2000-2016 | °C | GlobPermafrost doi: 10.1594/PANGAEA.888600 |
| Mean Annual Ground Temperature σ 2000-2016 | °C | GlobPermafrost doi: : 10.1594/PANGAEA.888600 |
| Permafrost Probability 2000-2016 | - | GlobPermafrost doi: : 10.1594/PANGAEA.888600 |

660    The set of 18 variables used in our study was carefully selected to capture the broad environmental conditions that are the important drivers of hydro-biogeochemical processes, and GHG fluxes, in Arctic ecosystems. The selected variables cover meteorological and bioclimatic conditions, soil properties, topographic, and permafrost conditions. Meteorological and bioclimatic conditions are primary drivers of vegetation, biological and ecological processes, and a similar selection of variables as chosen herein have been demonstrated to perform well for upscaling purposes in past published studies (Schimel

665    et al., 2007; Jung et al., 2009, 2011; Dengel et al., 2013; Knox et al., 2016, 2019; Jung et al., 2020; Malone et al., 2021). Complex microtopography is known to be an important driver of microclimate and vegetation in many parts of the Arctic,

and is represented here by the compound topographic index (CTI), a parameter designed to capture the impact of topography on hydrological processes. In addition to surface processes and vegetation, subsurface biogeochemical processes play an important role in high latitude Arctic ecosystems. Soil properties used for this study, including e.g. bulk density, or carbon and nitrogen contents, were selected to capture the heterogeneous subsurface conditions. Ecosystems in the vast Arctic region span continuous, discontinuous and sporadic permafrost conditions, and varying seasonal permafrost thaw conditions that regulate the GHG fluxes. Three permafrost related variables were therefore selected to reflect these heterogeneous conditions. In conclusion, even though not all of the 18 variables selected for our study are directly connected to variability in GHG fluxes, their combination is, to our knowledge, the best representation to capture the variability in environmental drivers that influence biogeochemical processes, and thus also the GHG fluxes across the Arctic.

**Appendix B**

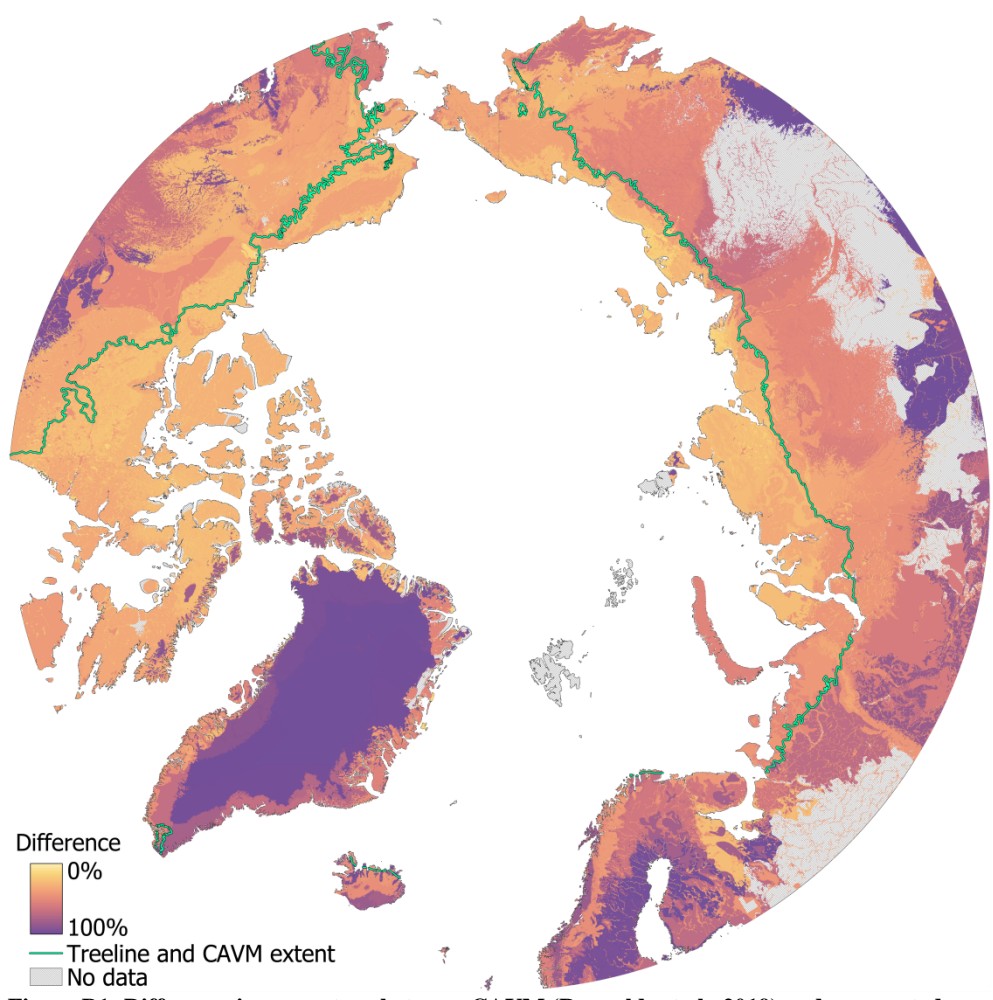

Difference
0%
100%
— Treeline and CAVM extent
No data

**Figure B1: Difference in percentage between CAVM (Raynolds et al., 2019) and aggregated ecoregions. Where both maps intersect**
680 **in the CAVM domain, we see the best fit, which is to be expected as there is no reference outside this domain.**

100 Ecoregions

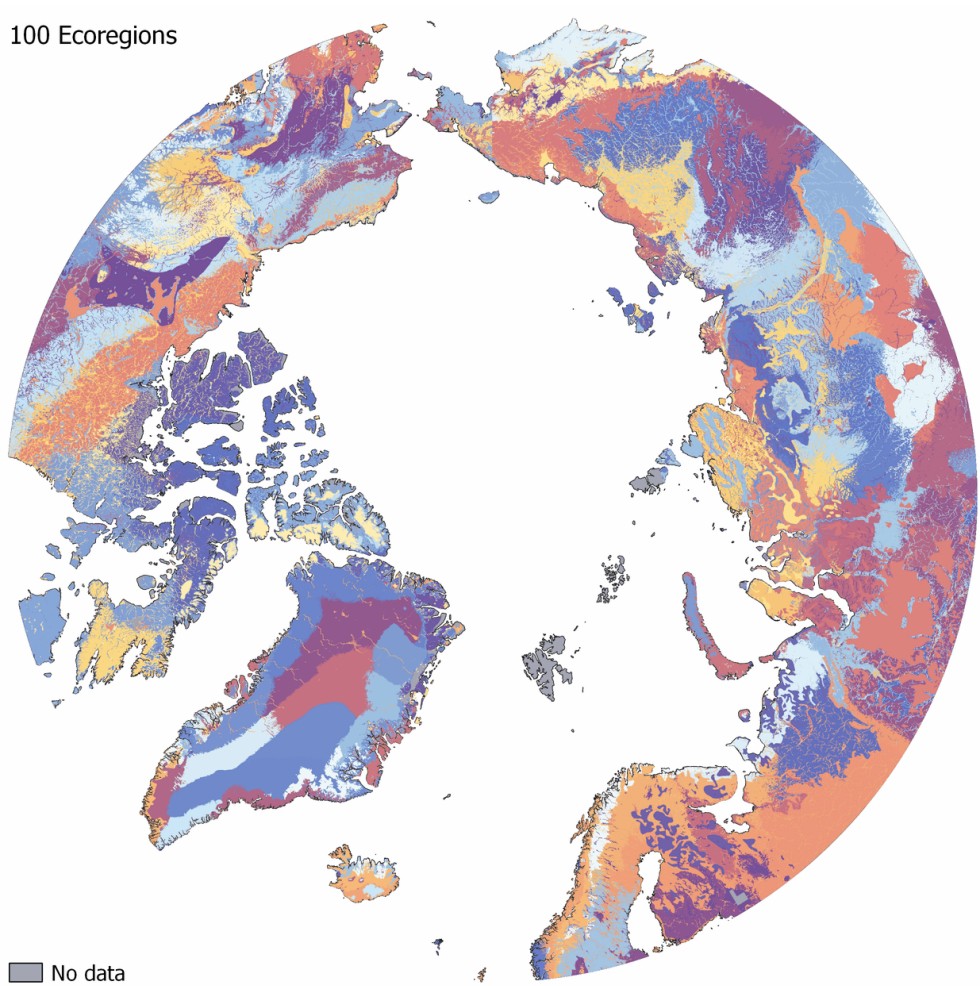

No data

**Figure B2: Overview of the 100 ecoregions as detailed in section 2.3.**


**Appendix C**

Active

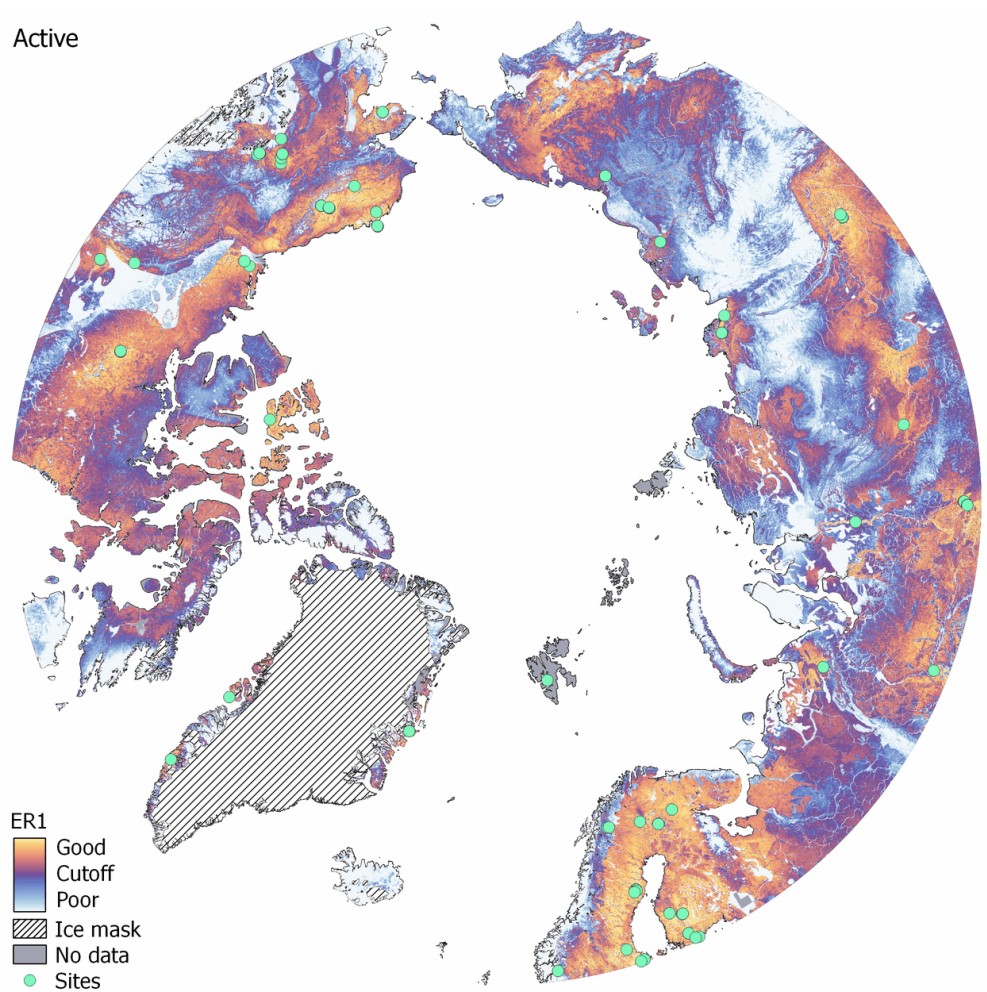

ER1
Good
Cutoff
Poor
Ice mask
No data
Sites

**Figure C1: Representativeness of the *Active* subsets. The center value of the color spectrum equals the ER1 cutoff (i.e. the 75th percentile of representativeness values of ecoregions with at least one site), thus warm/yellow colors match represented regions whereas cold/blue colors do not meet this criterion.**


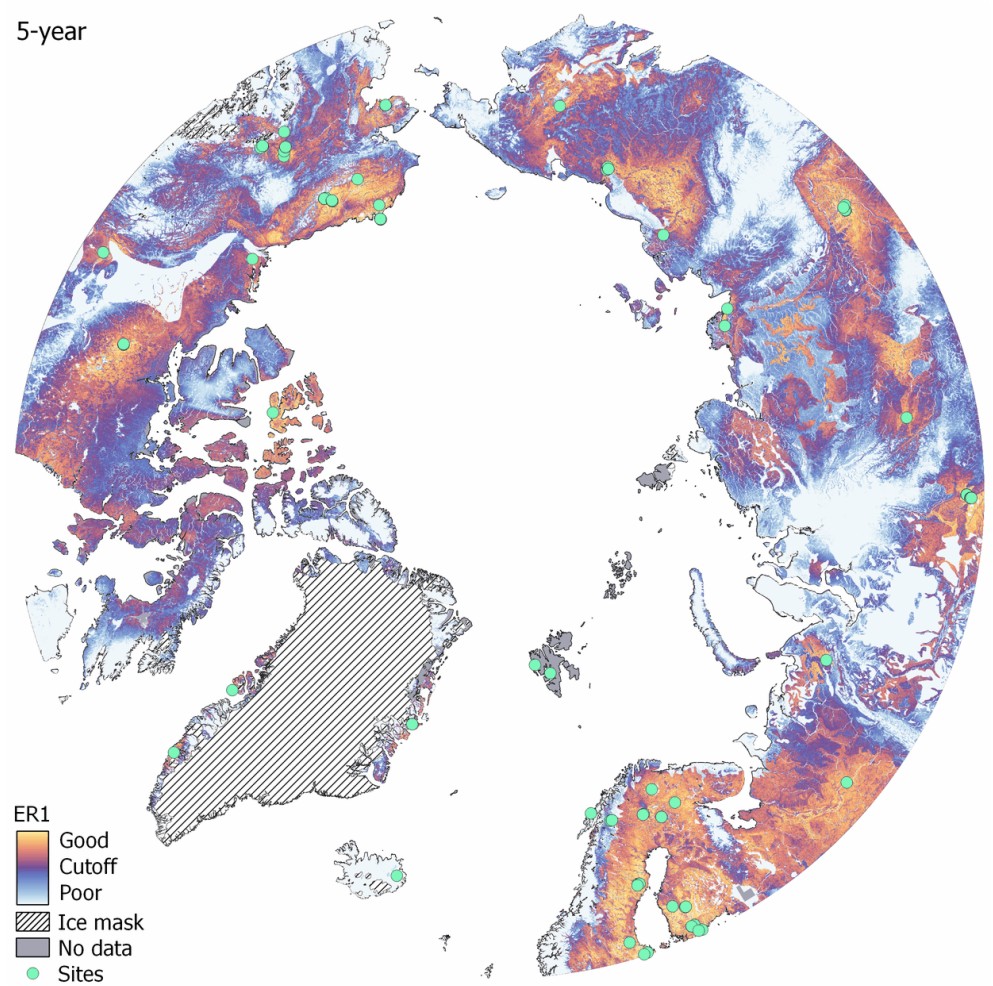

**5-year**

**ER1**
Good
Cutoff
Poor
Ice mask
No data
● Sites

**Figure C2: Representativeness of the *5-year* subsets. The center value of the color spectrum equals the ER1 cutoff (i.e. the 75[th] percentile of representativeness values of ecoregions with at least one site), thus warm/yellow colors match represented regions whereas cold/blue colors do not meet this criterion.**

Active

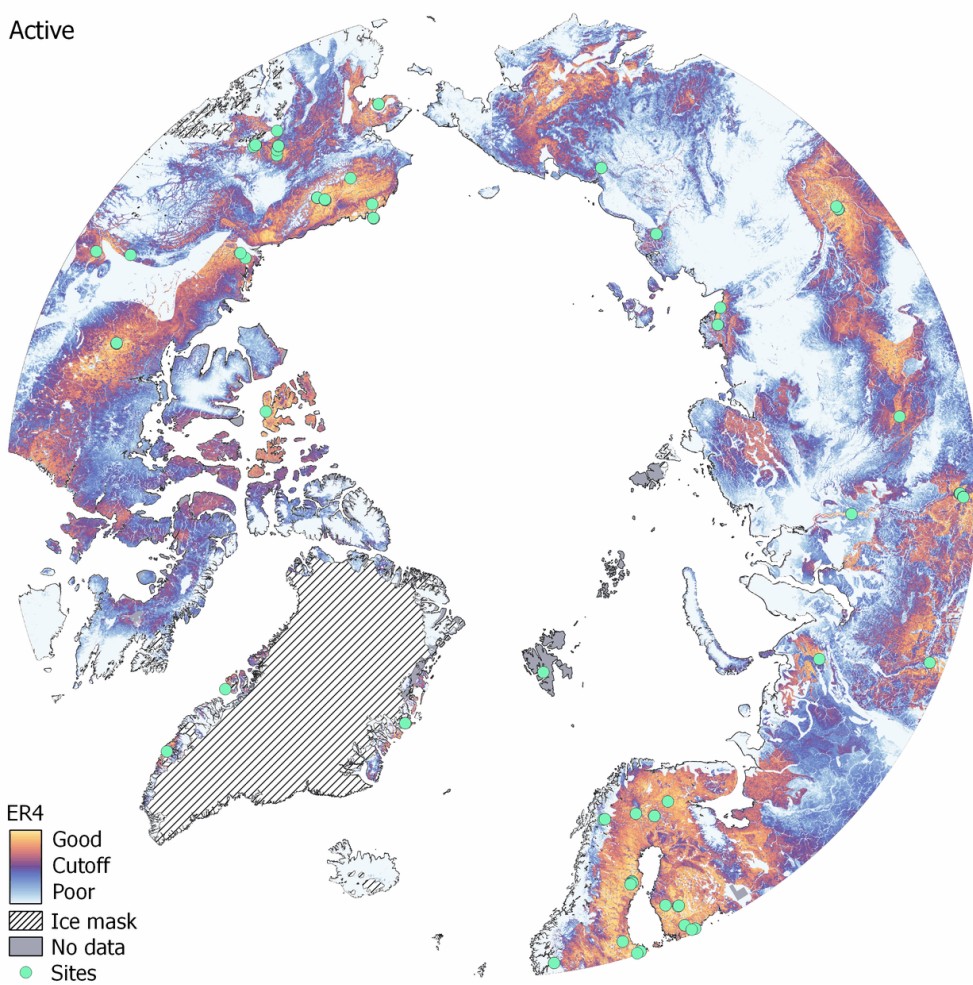

**Figure D1: Representativeness of the *Active* subset of sites using the ER4 cutoff as a center value of the color spectrum (i.e. the 75[th] percentile of representativeness values of ecoregions with at least four sites). Warm/yellow colors indicate regions that fall within the cutoff, i.e. represented regions, whereas cold/blue colors do not meet this criterion. The representativeness values underlying this map are identical to those used in Fig. C1 above, but due to the stricter ER4 quality criteria, the size of those domains that falls within this cutoff is lower. However this region can realistically be upscaled from the existing network .**


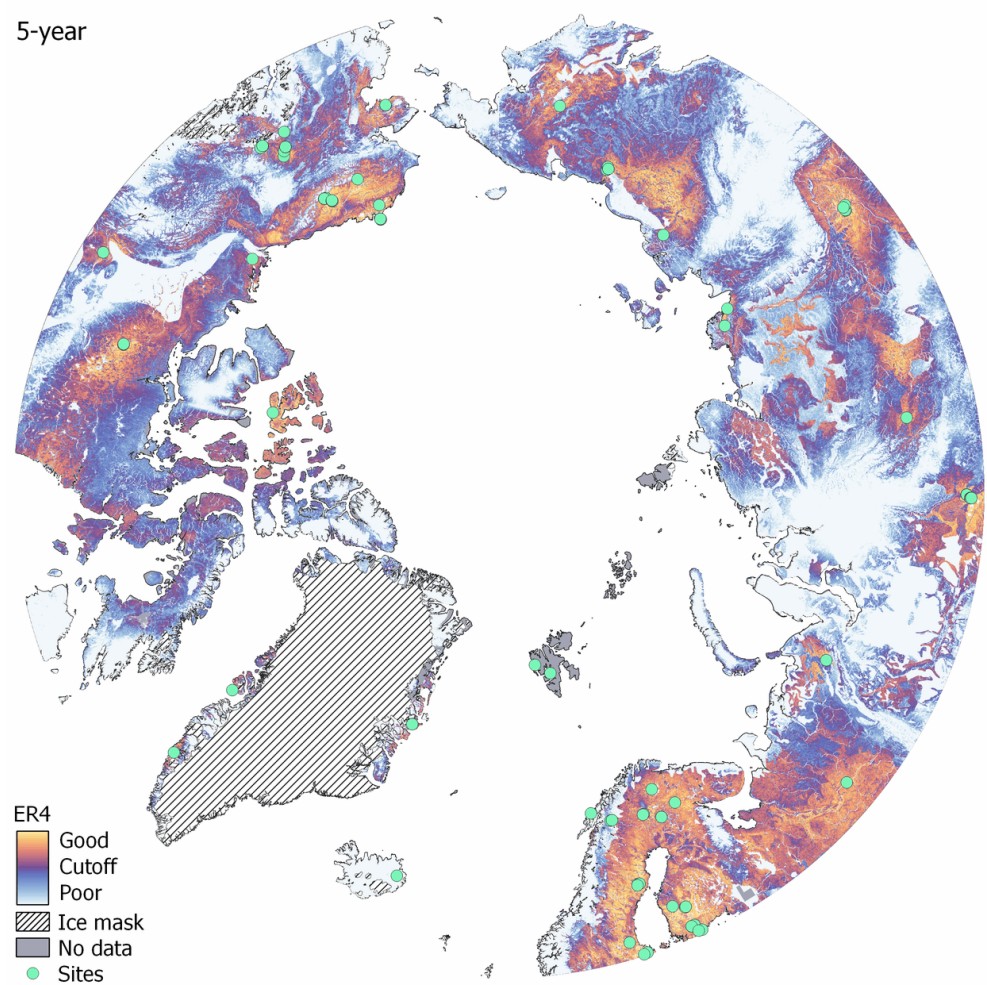

**Figure D2: Representativeness of the *5-year* subset of sites using the ER4 cutoff as a center value of the color spectrum (i.e. the 75th percentile of representativeness values of ecoregions with at least four sites). Warm/yellow colors indicate regions that fall within the cutoff, i.e. represented regions, whereas cold/blue colors do not meet this criterion. The representativeness values underlying this map are identical to those used in Fig. C1 above, but due to the stricter ER4 quality criteria, the size of those domains that falls within this cutoff is lower. However this region can realistically be upscaled from the existing network .**



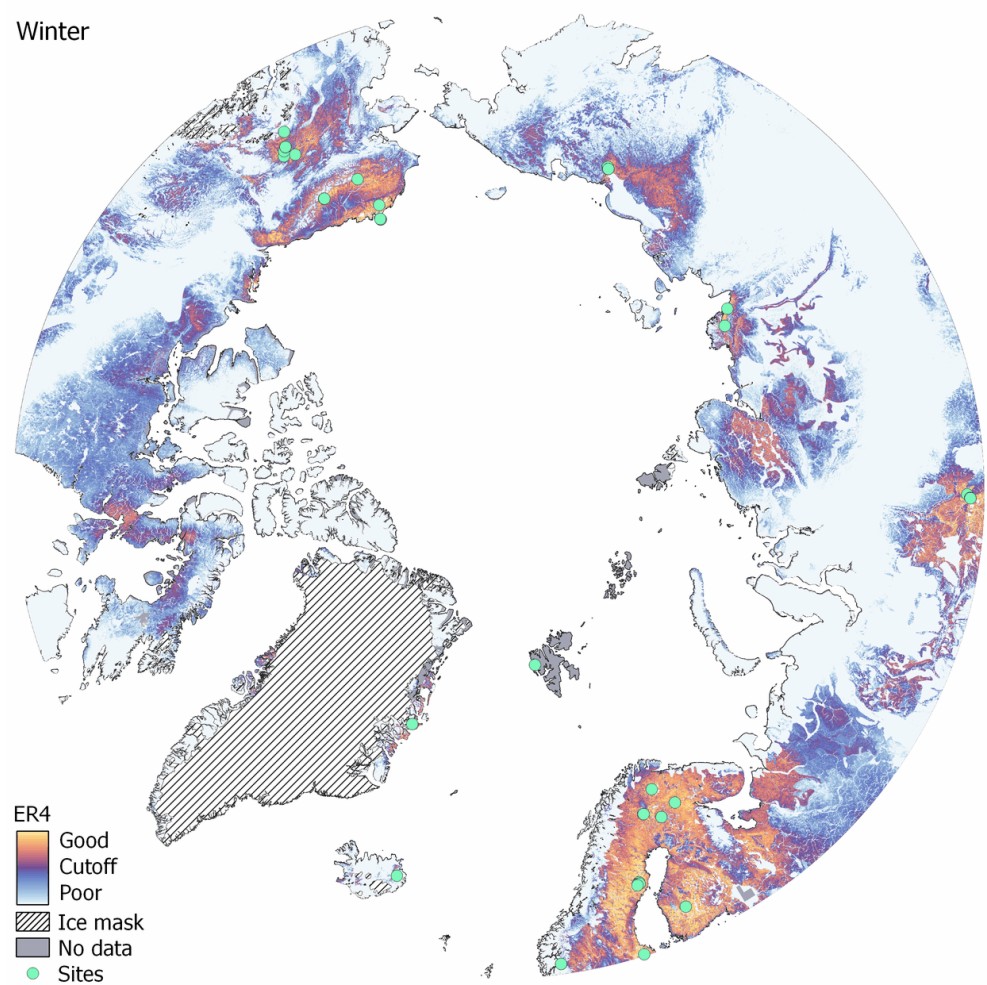

Winter

ER4
Good
Cutoff
Poor
Ice mask
No data
Sites

**Figure D3: Representativeness of the *Winter* subset of sites using the ER4 cutoff as a center value of the color spectrum (i.e. the 75th percentile of representativeness values of ecoregions with at least four sites). Warm/yellow colors indicate regions that fall within the cutoff, i.e. represented regions, whereas cold/blue colors do not meet this criterion. The representativeness values underlying this map are identical to those used in Fig. C1 above, but due to the stricter ER4 quality criteria, the size of those domains that falls within this cutoff is lower. However this region can realistically be upscaled from the existing network .**


CH₄

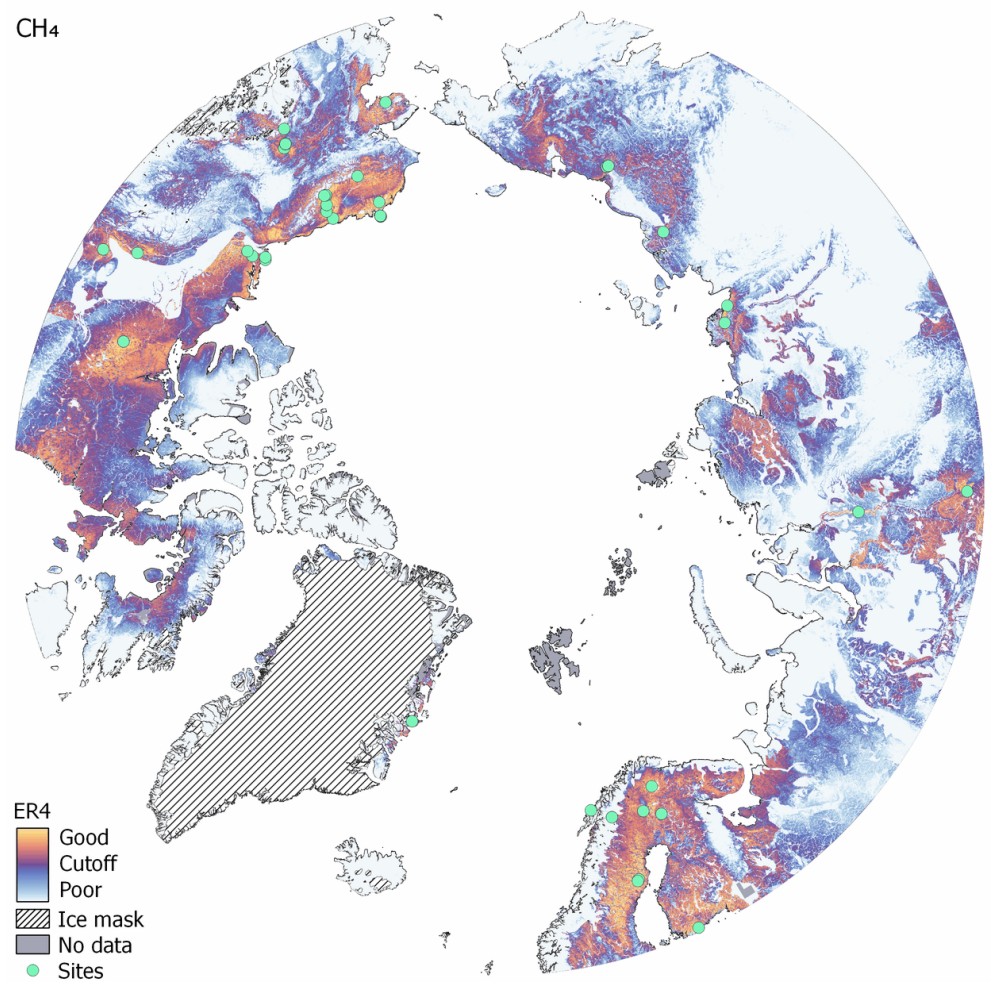

ER4
Good
Cutoff
Poor
Ice mask
No data
● Sites

**Figure D4: Representativeness of the *CH₄* subset of sites using the ER4 cutoff as a center value of the color spectrum (i.e. the 75ᵗʰ**
**percentile of representativeness values of ecoregions with at least four sites). Warm/yellow colors indicate regions that fall within**
**the cutoff, i.e. represented regions, whereas cold/blue colors do not meet this criterion. The representativeness values underlying**
**this map are identical to those used in Fig. C1 above, but due to the stricter ER4 quality criteria, the size of those domains that**
**falls within this cutoff is lower. However this region can realistically be upscaled from the existing network .**

Winter-CH₄

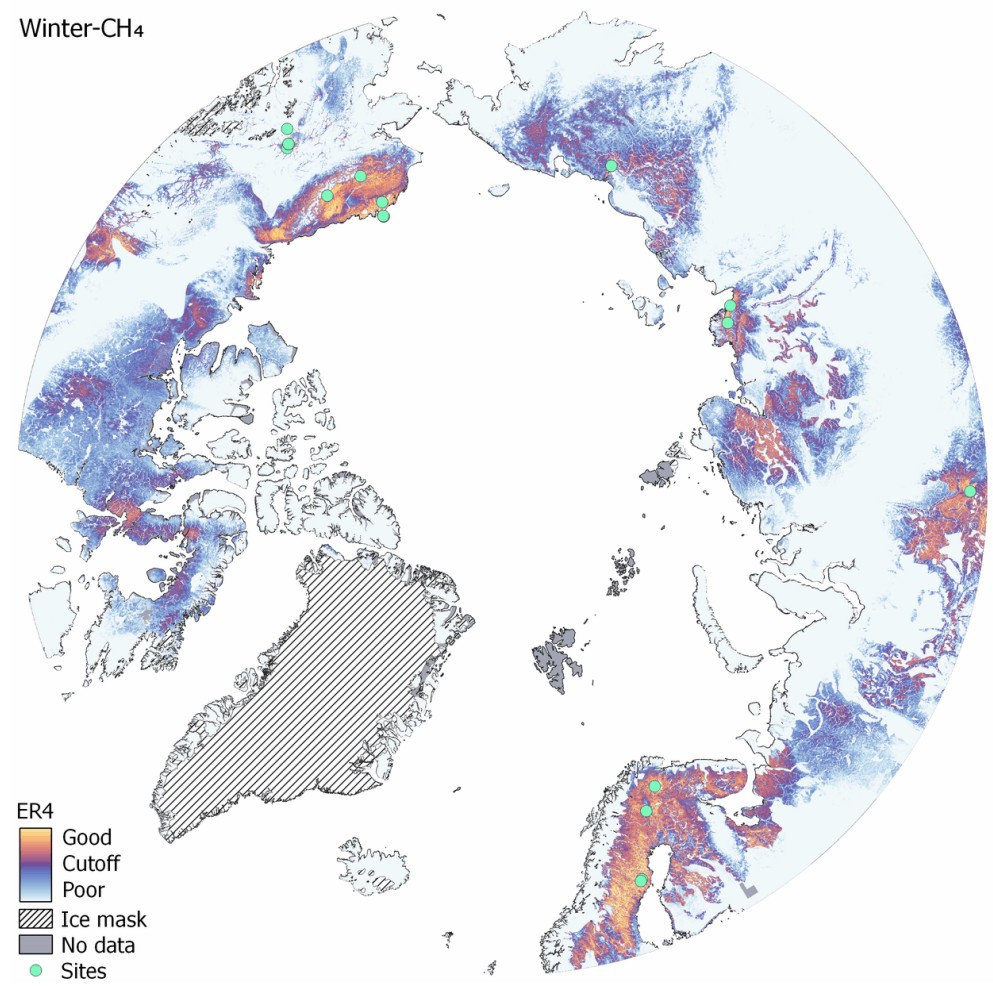

ER4
Good
Cutoff
Poor
Ice mask
No data
Sites

**Figure D5: Representativeness of the *Winter-CH₄* subset of sites using the ER4 cutoff as a center value of the color spectrum (i.e. the 75ᵗʰ percentile of representativeness values of ecoregions with at least four sites). Warm/yellow colors indicate regions that fall within the cutoff, i.e. represented regions, whereas cold/blue colors do not meet this criterion. The representativeness values underlying this map are identical to those used in Fig. C1 above, but due to the stricter ER4 quality criteria, the size of those domains that falls within this cutoff is lower. However this region can realistically be upscaled from the existing network .**



## Appendix E

**Table E1: selected new site locations for the *All* scenario. And upgrade locations for the *Winter*, *CH₄* and *Winter-CH₄* subsets. For each subset, locations are ordered from starting with the best improvement to the network.**

| Subset | Location | Site ID | Latitude | Longitude |
|--------|----------|---------|----------|-----------|
| *All* | Zhilinda | | 70.13 | 114.00 |
| | Omolon | | 65.25 | 160.50 |
| | Susuman | | 62.78 | 148.17 |
| | Olyokminsk | | 60.50 | 120.39 |
| | Iqaluit | | 63.75 | -68.50 |
| | Noyabrsk | | 63.17 | 75.62 |
| | Ust Nera | | 64.57 | 143.20 |
| | Agapa | | 71.45 | 89.25 |
| | Khorgo | | 73.48 | 113.63 |
| | Udachnyy | | 66.42 | 112.40 |
| *Winter* | Tura | RU-TUR | 64.21 | 100.46 |
| | Chukotka | RU-CUK | 65.59 | 171.05 |
| | Cape Bounty | - | 74.92 | -109.56 |
| | Mukhrino | - | 60.90 | 68.70 |
| | Daring Lake | CA-DL1 | 64.86 | -111.57 |
| | Yakutsk-Pine | RU-YPF | 62.24 | 129.65 |
| | Smith Creek | CA-SMC | 63.15 | -123.25 |
| | Neleger | RU-NEL | 62.08 | 129.75 |
| | Ust Pojeg | RU-UPO | 61.93 | 50.23 |
| | Seida | RU-VRK | 67.05 | 62.94 |
| *CH₄* | Tura | RU-TUR | 64.21 | 100.46 |
| | Cape Bounty | - | 74.92 | -109.56 |
| | Mukhrino | - | 60.90 | 68.70 |
| | Ust Pojeg | RU-UPO | 61.93 | 50.23 |
| | Chukotka | RU-CUK | 65.59 | 171.05 |
| | Yakutsk-Pine | RU-YPF | 62.24 | 129.65 |
| | Neleger | RU-NEL | 62.08 | 129.75 |
| | Seida | RU-VRK | 67.05 | 62.94 |
| | Varrio | FI-VAR | 67.75 | 29.61 |
| | Spasskaya Pad | RU-SKP | 62.26 | 129.17 |
| *Winter-CH₄* | Tura | RU-TUR | 64.21 | 100.46 |
| | Ust Pojeg | RU-UPO | 61.93 | 50.23 |
| | Chukotka | RU-CUK | 65.59 | 171.05 |
| | Cape Bounty | - | 74.92 | -109.56 |
| | Mukhrino | - | 60.90 | 68.70 |
| | Daring Lake | CA-DL1 | 64.86 | -111.57 |
| | Yakutsk-Pine | RU-YPF | 62.24 | 129.65 |
| | Council | - | 64.84 | -163.71 |
| | Smith Creek | CA-SMC | 63.15 | -123.25 |
| | Spasskaya Pad | RU-SKP | 62.26 | 129.17 |


**Code/Data availability**

The EC site list and metadata can be accesses at http://cosima.nceas.ucsb.edu/carbon-flux-sites. The current state of the online database will deviate slightly from the version used in this paper, since we continuously work in data updates provided by site PIs and encourage PIs of new sites to contact us in the future too. For reference, a version of the database reflecting the state that was used to produce results summarized in this manuscript has been retained, and is available on request.

Individual representativeness maps for each site will be made openly available. These can be used to assess the impact of individual sites or reproduces the network maps as shown in this paper.

The Department of Energy will provide public access to these results of federally sponsored research in accordance with the DOE Public Access Plan (http://energy.gov/downloads/doe-public-access-plan).

**Author contribution**

MG and MP conceptualized the study. MP, MM, AV and MG collected EC metadata, MP conducted the survey and analysed the EC metadata. MP and GC developed the online Arctic GHG site mapping tool. JK and FH implemented the representativeness, mapcurves and clustering algorithms. MP implemented the optimization algorithms, and analysed all results. MP wrote the original draft, with notable reviews and edits from JK, MM, ES, AV and MG.

**Competing interests**

The authors declare that they have no conflict of interest.

**Acknowledgements**

This work was supported by the Max-Planck Society, and through funding by the European Commission (INTAROS project, H2020-BG-09-2016, Grant Agreement No. 727890). JK and FH were supported by The Next-Generation Ecosystem Experiments (NGEE Arctic) and the Reducing Uncertainties in Biogeochemical Interactions through Synthesis and Computation Scientific Focus Area (RUBISCO SFA), which are sponsored by the Office of Biological and Environmental Research in the DOE Office of Science. We would like to acknowledge FLUXNET and the regional databases, e.g. Ameriflux, AsiaFlux, European Fluxes Database Cluster, ICOS, NEON for making flux data readily available. We would like to thank the Arctic Data Center for hosting the online Arctic GHG site mapping tool. Furthermore, we would like to extend a special thanks to all colleagues that either participated in our survey or contacted us directly about their high latitude site.

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
