# Peer review of "Representativeness assessment of the pan-Arctic eddy-covariance site network, and optimized future enhancements"

_Biogeosciences, 2021_

## Author Comment (AC1)

General Comments:

This paper synthesizes available eddy covariance data in the Arctic and attempts to identify areas which are well and not well represented by the current distribution of eddy covariance stations. The topic is interesting and important, the paper is generally well written, and the analysis appears sound though the spatial representativeness assessment lies well outside my own competence. A major source of impact and novelty in the present paper is the method used to identify specific new locations/upgrades to existing locations which would result in the greatest relative increase in biome representativeness. This advance is particularly valuable since it could greatly improve the effectiveness of strategic research planning for future flux sampling efforts. I have only fairly minor suggestions for improvements (see below).

Specific Comment:

All figures: It is difficult to distinguish the "no data" and poorly represented areas with the current color scheme. Also, it may be a good idea to explicitly exclude areas with permanent ice (ie: much of Greenland), unless you think these areas should be represented with EC data?

We will change the color scheme for the no data and ice sheet regions. The final figures will be provided in higher resolution, which will also aid in distinguishing between the different cases.

Lines 136-139) I miss a bit more detailed discussion and justification in the intro, results and discussion of the broader implications/considerations of the 18 variables chosen to represent environmental variability. How/why were these particular variables chosen? Are there any other significant variables that could have been interesting to include? How might the variables selected in turn impact your estimates of spatial variability in representativeness? For example, are there some important variables which remain poorly represented even in Alaska/Fennoscandia? How were the variables combined together to create a single metric of environmental variability? Many of the 18 variables seem likely they would be strongly autocorrelated with each other, so any procedure that treats all individual variables as "equal" in weight to each other may be flawed…

The combination of these variables into a single metric is discussed in the first paragraph of methods section 2.2. The representativeness value is the Euclidian distance in n-dimensional space between each location in our domain and the closest EC site. Here the n-dimensional space consists of the normalized bioclimatic variables.

This issue of the 18 variables was also raised by reviewer 2, and in the response to his comments we discuss it in detail in our answer A1. We copied this response below.

We agree that indeed we are estimating the variability of these 18 variables and not the GHG fluxes themselves. And while we do not discuss this limitation in 4.3, we did in 4.2:
"*However, the assessment of specific fluxes provided by the eddy covariance tower network based on these data layers must largely remain qualitative, since no clear quantitative linkage between the bioclimatic controls and the fluxes for $CO_2$ and $CH_4$ can be considered. In other words, assigning equal weights to all 18 data layers allows us to assess a general level of*

*similarity between pixels within the domain, but does not necessarily reflect how strongly potential differences will influence greenhouse gas flux rates (see e.g. Tramontana et al.(2020))"*

The application of this type of method of using ecological land cover classifications for EC network analysis has been successfully demonstrated in earlier references (e.g. Hargrove and Hoffman, 2004; Hoffman et al., 2013). However, we agree with the reviewer that this core assumption to our study needs to be presented more prominently in the text. Therefore, we plan to change the second last paragraph of the introduction to:

*"Building on a study by Hoffman et al. (2013) that presented an analysis of the Alaskan EC network, in this study we will provide a first in-depth evaluation of the current and past pan-Arctic EC flux observation infrastructure. Our method uses quantitative multivariate clustering which has many uses from creating maps of geological regions (Harff and Davis, 1990), to watershed delineation (Hessburg et al., 2000) and ecoregion classification (Zhou et al., 2003). Hargrove and Hoffman (2004) give an extensive overview of these applications, which are all based on the concept of mapping normalized ecosystem variables such as topography, precipitation and temperature in an n-dimensional data space, using one axis for each variable. The closer two points are in this variable space, the more alike they are, and the more likely they are to be classified as belonging to the same ecoregion when clustered by a k-means algorithm. Thus, the distance can be interpreted as a metric of variability. Aiming at assessing the representativeness of the EC network in the US, Hargrove and Hoffman (2004) then calculated the distances between each constructed ecoregion without an EC site to the closest ecoregion with an EC site. Hoffman (2013) later extended this method to map the Alaska EC network, this time, instead of aggregating the distances between ecoregions, calculating the distance between each pixel in the map and the closest EC site. This approach thus preserves the fine scale variability that is lost when aggregating to the ecoregion level.. In our implementation we will also perform this analysis on an individual pixel scale."*

Hill et al. (2017)explain the statistical requirements for upscaling fluxes from site levels to the ecosystem level. No actual fluxes are required with their typical example, just the ecoregion/ecosystems and the number of towers present. We will add according information to the last paragraph (more details in the next answer A2):

*"Our analysis aims to quantify representativeness in the pan-Arctic domain based on this similarity in key ecosystem characteristics of any location in our domain to those of the EC sites. We further use the analysis by Hill et al. (2017) on the statistical power of EC systems to put these representatives measures into perspective regarding the general potential to upscale fluxes from sparse EC networks. Moreover, we use the results from the representativeness analyses to identify the most suitable locations for new observation sites, and upgrades to existing infrastructure, that would optimally enhance the performance of the Arctic EC network as a whole. Finally, this manuscript and its corresponding online tool aim at providing an easily accessible source of information on Arctic flux monitoring infrastructure and literature for scientists working on the carbon cycle."*

Regarding the selection of variables, and to which degree they are relevant for EC flux network, we will add the following explanation and addition to the paper as part of the new third paragraph of the introduction:

*"Knowing the current and past spatiotemporal distribution of EC sites is not enough to fully understand to which degree this network represents the Arctic domain. The reason for this is that EC towers have a field of view that typically does not extend further than a kilometer from the tower, often less (Leclerc and Thurtell, 1990; Horst and Weil, 1992; Schmid, 1994, 2002; Vesala et al., 2008). Accordingly, with currently about 120 terrestrial EC towers situated within the Arctic domain, only a very small fraction of the region gets directly observed, while most of its expanse remains unsampled. Larger footprints would not solve this problem, as the greater heterogeneity would still be hard to capture. Meteorology, vegetation, above/below ground conditions, and topography are critical drivers of hydrological and biogeochemical processes at landscape scale and of GHG fluxes, and their variability across the Arctic therefore also causes variability in flux rates. For upscaling purposes (i.e., when fluxes are predicted over larger areas), typically a tower is held as representative for the ecosystem and the region where it is stationed (Desai, 2010; Jung et al., 2011; Xiao et al., 2012; Chu et al., 2021); however, except when using a very coarse classification of ecosystem types, the existing EC network still cannot cover all ecosystems across the Arctic, and a courser classification would increase heterogeneity within the ecosystem and reduce the representation within the ecosystem. Still, a number of published studies have successfully demonstrated the effectiveness of using meteorological and environmental variables as explanatory variables for estimating GHG fluxes at regional to global scales (e.g. Jung et al., 2020; Knox et al., 2019)."*

To provide further insights into the rationale behind choosing the 18 bio-climatic variables used in our study, we plan to add the following text to Appendix 1:

*"The set of 18 variables used in our study was carefully selected to capture the broad environmental conditions that are the important drivers of hydro-biogeochemical processes, and GHG fluxes, in the Arctic ecosystem. The selected variables cover meteorological and bioclimatic conditions, soil properties, topographic, and permafrost conditions. Meteorological and bioclimatic conditions are primary drivers of vegetation, biological and ecological processes, and a similar selection of variables as chosen herein have been demonstrated to perform well for upscaling purposes in past published studies (Schimel et al., 2007; Jung et al., 2009, 2011; Dengel et al., 2013; Knox et al., 2016, 2019; Jung et al., 2020; Malone et al., 2021). Complex microtopography is known to be an important driver of microclimate and vegetation in many parts of the Arctic, and is represented here by the compound topographic index (CTI), a parameter designed to capture the impact of topography on hydrological processes. In addition to surface processes and vegetation, subsurface biogeochemical processes play an important role in high latitude Arctic ecosystems. Soil properties used for this study, including e.g. bulk density, or carbon and nitrogen contents, were selected to capture the heterogeneous subsurface conditions. Ecosystems in the vast Arctic region span continuous, discontinuous and sporadic permafrost conditions, and varying seasonal permafrost thaw conditions that regulate the GHG fluxes. Three permafrost related variables were therefore selected to reflect these heterogeneous conditions. In conclusion, even though not all of the 18 variables selected for our study are directly connected to variability in GHG fluxes, their combination is, to our knowledge, the best*

*representation to capture the variability in environmental drivers that influence biogeochemical processes, and thus also the GHG fluxes across the Arctic."*

A geostatistical method, while certainly a viable approach, was not performed since it requires a large dataset of EC fluxes on top of the raster datasets of bio-climatic variables. Collecting the actual flux data for the 120+ EC sites analyzed within our study was beyond the scope of this project, our concept builds on their metadata alone. We fully agree with the reviewer that the geostatistical approach would provide more certainty in its selection (or weighing) of variables; however, we do not see a practical way of building the required EC database with reasonable expenses. We therefore decided to apply the distance-based approach promoted by Hoffman et al. (2013), and accept (and discuss) the extrapolation uncertainty that comes with weighing all selected bio-climatic parameters equally.

Lines 509-501) Its probably true that more EC sites within an ecosystem would create greater certainty for that ecosystem but the key question is would this result in a greater % improvement in overall biome representativeness compared to installing an EC site following your optimized site selection protocol? I assume not, but your approach should be able to resolve this.

The answer to that question depends on the scale of analysis. When looking at variability within pixels, we clearly cannot differentiate with this method; however, having multiple towers located in pixels with very similar characteristics can affect the number of sites outside that pixel that fall within the ER5 metric. On an ecoregion scale, as described in lines 419-425 and 544-549 adding an additional site to ecoregions that already contain a site in certain cases can indeed be slightly more beneficial for the network-wide representativeness than adding a site to a new region. The decision where to place new sites also depends a bit on the objective, i.e. whether the representativeness, ER1 or ER5 metric is supposed to be the target. We see in our data that the representativeness and fraction of area in ER1 and ER5 are all highly correlated. This means that if we improve the overall representativeness as is the target in the current optimization setup, we will also increase the ER5 value which is a measure of high confidence in similar ecoregions. For future research it would be interesting to also optimize directly for the ER5 (or similar) metric and compare this to the optimization of the representativeness as a whole.

---

## Author Comment (AC2)

The manuscript is centered around the question on how representative a network of greenhouse gas (GHG) flux observation sites in the Arctic is. This is a valid and relevant question and the authors give several examples of research questions for which representativeness of flux sites for larger entities is an issue. The analysis can guide decisions where to optimally place future investment in additional field stations and into upgrading existing stations. With the metadata base on arctic flux stations, including those that are not yet part of any flux network, and the online data base tool to make this information generally accessible, the work provides a service to the arctic flux community and the users of its products.

The manuscript is well written and the presentations of the results work well.

A specific limitation of the manuscript is that the largely theoretical approach builds on a number of untested propositions, assumptions and hypotheses (see below), and thus the results remain speculative, i.e. mere results of the input. This sounds worse than it is, because from a practical point of view there might not be much to be done about it and this method may encompass one of the better available approaches, if one 'believes' at all in the possibility to extrapolate from as little as 120 + stations to an entire global zone (see section 4.3). Well, technically it is possible, as important publications have demonstrated, and for this reason alone this work might be important, to understand the risk of doing so.

Since most of detailed comments below are related to the following main points, we answer to these 4 points here in detail, and number the answers (A1 to A4). In the responses to the detailed comments, we then refer back to these answers for the full explanation.

- The main scientific problem is to estimate representativeness from a sparse sample. The ideal statistical approach would be working with a luxuriously large sample size and then estimating the minimum sample size at any given spatial distribution to achieve a defined acceptable uncertainty of the extrapolated numbers, but this is not possible for sparse samples. In this situation, the Authors make an elegant move, to analyse representativeness from ample but different dataset that represent the spatial variability at high resolution (1 km^2). But strictly speaking, this approach estimates representativeness for the 18 variables and their combinations and not the GHG fluxes.
- Whether the assumption that the ecological land cover classification is valid also for the GHG fluxes, is neither tested nor shown, even questioned in their section 4.3). The whole results rest on this untested hypothesis that spatial patterns of GHG flux variability are the same or at least similar to the spatial variability of the 18 variables as aggregated by their approach. A geostatistical rigorous approach would start with showing the quantitative relationship between the site factor proxies (the 18 variables) and the observed fluxes and than use this to determine and minimize the upscaling error. This was not attempted.

A1.

We agree that indeed we are estimating the variability of these 18 variables and not the GHG fluxes themselves. And while we do not discuss this limitation in 4.3, we did in 4.2:
*"However, the assessment of specific fluxes provided by the eddy covariance tower network*

*based on these data layers must largely remain qualitative, since no clear quantitative linkage between the bioclimatic controls and the fluxes for $CO_2$ and $CH_4$ can be considered. In other words, assigning equal weights to all 18 data layers allows us to assess a general level of similarity between pixels within the domain, but does not necessarily reflect how strongly potential differences will influence greenhouse gas flux rates (see e.g. Tramontana et al.(2020))"*

The application of this type of method of using ecological land cover classifications for EC network analysis has been successfully demonstrated in earlier references (e.g. Hargrove and Hoffman, 2004; Hoffman et al., 2013). However, we agree with the reviewer that this core assumption to our study needs to be presented more prominently in the text. Therefore, we plan to change the second last paragraph of the introduction to:

*"Building on a study by Hoffman et al. (2013) that presented an analysis of the Alaskan EC network, in this study we will provide a first in-depth evaluation of the current and past pan-Arctic EC flux observation infrastructure. Our method uses quantitative multivariate clustering which has many uses from creating maps of geological regions (Harff and Davis, 1990), to watershed delineation (Hessburg et al., 2000) and ecoregion classification (Zhou et al., 2003). Hargrove and Hoffman (2004) give an extensive overview of these applications, which are all based on the concept of mapping normalized ecosystem variables such as topography, precipitation and temperature in an n-dimensional data space, using one axis for each variable. The closer two points are in this variable space, the more alike they are, and the more likely they are to be classified as belonging to the same ecoregion when clustered by a k-means algorithm. Thus, the distance can be interpreted as a metric of variability. Aiming at assessing the representativeness of the EC network in the US, Hargrove and Hoffman (2004) then calculated the distances between each constructed ecoregion without an EC site to the closest ecoregion with an EC site. Hoffman (2013) later extended this method to map the Alaska EC network, this time, instead of aggregating the distances between ecoregions, calculating the distance between each pixel in the map and the closest EC site. This approach thus preserves the fine scale variability that is lost when aggregating to the ecoregion level.. In our implementation we will also perform this analysis on an individual pixel scale."*

Hill et al. (2017) explain the statistical requirements for upscaling fluxes from site levels to the ecosystem level. No actual fluxes are required with their typical example, just the ecoregion/ecosystems and the number of towers present. We will add according information to the last paragraph (more details in the next answer A2):

*"Our analysis aims to quantify representativeness in the pan-Arctic domain based on this similarity in key ecosystem characteristics of any location in our domain to those of the EC sites. We further use the analysis by Hill et al. (2017) on the statistical power of EC systems to put these representatives measures into perspective regarding the general potential to upscale fluxes from sparse EC networks. Moreover, we use the results from the representativeness analyses to identify the most suitable locations for new observation sites, and upgrades to existing infrastructure, that would optimally enhance the performance of the Arctic EC network as a whole. Finally, this manuscript and its corresponding online tool aim at providing an easily accessible source of information on Arctic flux monitoring infrastructure and literature for scientists working on the carbon cycle."*

Regarding the selection of variables, and to which degree they are relevant for EC flux network, we will add the following explanation and addition to the paper as part of the new third paragraph of the introduction:

*"Knowing the current and past spatiotemporal distribution of EC sites is not enough to fully understand to which degree this network represents the Arctic domain. The reason for this is that EC towers have a field of view that typically does not extend further than a kilometer from the tower, often less (Leclerc and Thurtell, 1990; Horst and Weil, 1992; Schmid, 1994, 2002; Vesala et al., 2008). Accordingly, with currently about 120 terrestrial EC towers situated within the Arctic domain, only a very small fraction of the region gets directly observed, while most of its expanse remains unsampled. Larger footprints would not solve this problem, as the greater heterogeneity would still be hard to capture. Meteorology, vegetation, above/below ground conditions, and topography are critical drivers of hydrological and biogeochemical processes at landscape scale and of GHG fluxes, and their variability across the Arctic therefore also causes variability in flux rates. For upscaling purposes (i.e., when fluxes are predicted over larger areas), typically a tower is held as representative for the ecosystem and the region where it is stationed (Desai, 2010; Jung et al., 2011; Xiao et al., 2012; Chu et al., 2021); however, except when using a very coarse classification of ecosystem types, the existing EC network still cannot cover all ecosystems across the Arctic, and a courser classification would increase heterogeneity within the ecosystem and reduce the representation within the ecosystem. Still, a number of published studies have successfully demonstrated the effectiveness of using meteorological and environmental variables as explanatory variables for estimating GHG fluxes at regional to global scales (e.g. Jung et al., 2020; Knox et al., 2019)."*

To provide further insights into the rationale behind choosing the 18 bio-climatic variables used in our study, we plan to add the following text to Appendix 1:

*"The set of 18 variables used in our study was carefully selected to capture the broad environmental conditions that are the important drivers of hydro-biogeochemical processes, and GHG fluxes, in the Arctic ecosystem. The selected variables cover meteorological and bioclimatic conditions, soil properties, topographic, and permafrost conditions. Meteorological and bioclimatic conditions are primary drivers of vegetation, biological and ecological processes, and a similar selection of variables as chosen herein have been demonstrated to perform well for upscaling purposes in past published studies (Schimel et al., 2007; Jung et al., 2009, 2011; Dengel et al., 2013; Knox et al., 2016, 2019; Jung et al., 2020; Malone et al., 2021). Complex microtopography is known to be an important driver of microclimate and vegetation in many parts of the Arctic, and is represented here by the compound topographic index (CTI), a parameter designed to capture the impact of topography on hydrological processes. In addition to surface processes and vegetation, subsurface biogeochemical processes play an important role in high latitude Arctic ecosystems. Soil properties used for this study, including e.g. bulk density, or carbon and nitrogen contents, were selected to capture the heterogeneous subsurface conditions. Ecosystems in the vast Arctic region span continuous, discontinuous and sporadic permafrost conditions, and varying seasonal permafrost thaw conditions that regulate the GHG fluxes. Three permafrost related variables were therefore selected to reflect these heterogeneous conditions. In conclusion, even though not all of the 18 variables selected for our study are directly connected to variability in GHG fluxes, their combination is, to our knowledge, the best*

*representation to capture the variability in environmental drivers that influence biogeochemical processes, and thus also the GHG fluxes across the Arctic."*

A geostatistical method, while certainly a viable approach, was not performed since it requires a large dataset of EC fluxes on top of the raster datasets of bio-climatic variables. Collecting the actual flux data for the 120+ EC sites analyzed within our study was beyond the scope of this project, our concept builds on their metadata alone. We fully agree with the reviewer that the geostatistical approach would provide more certainty in its selection (or weighing) of variables; however, we do not see a practical way of building the required EC database with reasonable expenses. We therefore decided to apply the distance-based approach promoted by Hoffman et al. (2013), and accept (and discuss) the extrapolation uncertainty that comes with weighing all selected bio-climatic parameters equally.

- Using the *absolute* term "well represented" for the highest category of representativeness can indicate that the number, time series lengths and completeness of EC data is large enough to estimate the average flux of an ecoregion with sufficient certainty. But in fact this has not been shown. As it looks to me "well represented" means "well represented" relative to the upper 25% of the realized representativeness of EC stations and data (lines 164-169). But what does this mean for absolute representativeness? Is this amount of data sufficient for reasonable extrapolation/upscaling? I recommend using a relative term "higher relative representation", "lower relative representation" and discussion on, what this classification means for absolute representativeness. A possible result for the whole study might be that the network design should opt for increasing "relative representation" but this does not mean and does not be confused with sufficient representativeness for a certain purpose (e.g. upscaling).

A2.

While we performed a sensitivity test in our previous selection of ER5, we agree that it would be beneficial to be able to address more objectively the amount or quality of data which is required for proper characterization of an ecoregion for upscaling. Thus we propose to change ER5 to ER4 in line with the statistical analysis by Hill et al. (2017) which shows that, for a reasonably flat ecosystem, 4 towers are the minimum amount to properly characterize this.

We plan to change the last paragraph of Section 2.3 to explain this revised rationale:

*"To facilitate a quantitative assessment of network coverage, and put the results into context, we produced two derived metrics, subsequently labelled ER1 and ER4. Both include a threshold that allows separation of the study domain into areas that meet a defined requirement and those that do not, based on the representativeness score assigned to each pixel. We calculate these thresholds as the 75% percentile of the distribution of representativeness scores calculated for the All scenario described below, restricted to ecoregions that contain at least 1 site (ER1), or at least 4 sites (ER4), respectively. The ER1 metric represents the domain that is covered similarly*

*to an ecoregion with at least one EC tower, and can thus be interpreted as the fraction of the study area that the EC network provides basic information on. However, one tower typically does not provide enough information to reliably upscale fluxes to an entire ecoregion. Therefore, we added the ER4 metric to consider a minimum number of 4 towers required to reach a 0.95 statistical power to properly characterize an ecosystems EC fluxes (Hill et al., 2017). The ER4 metric can therefore be interpreted as showing that part of the study area for which the existing EC infrastructure allows upscaling of fluxes with a reasonable confidence. The requirement of 4 towers assumes relatively flat terrain (Baldocchi, 2003), while hilly or even rougher terrain would require at least 24 towers (Hill et al., 2017); however, since none of our ecoregions encompass this many towers we did not include this metric. The chosen cutoff at 75% generally follows studies which concluded that a perfect match between target conditions and observed conditions is unrealistic for EC sites, so that a deviation of 20-25% can still be considered 'homogeneous' (e.g. Göckede et al., 2008). In the presented study, applying this cutoff for each scenario as described below, the derived splitting point of representativeness values to meet the ER1 metric was calculated as 0.0089, while for the stricter ER4 metric this cutoff was 0.0063."*

Also, we plan to add this closer look at ER4 results to section 3.3:

*"It is of interest to note that as few as 5 site upgrades can double the size of the region of the Winter-CH$_4$ Network that meets the ER4 criteria. Between 15 to 25 new site additions to the All subset would be required to have a high (ER4) statistical confidence for upscaling that covers 50% of the Arctic terrestrial domain."*

This change requires that throughout the paper we will have to change ER5 to ER4 (and update its derived values). Instead of using the term '*well represented*' we plan to report the fractions of the domain that *fall within the metrics threshold* or *meet the metrics criteria*. This will change the interpretation. Regions that match the ER1 criteria will not be called well represented but being represented (by at least one tower with similar ecosystem characteristics). Regions that meet the ER4 criteria will be framed as regions for which upscaling will be possible with reasonable confidence.

- The number of ecoregions is subjectively chosen. This choice has a very high influence on the results. If that number of regions is very high, the representativeness of a given set of stations will be more extreme and vice versa. If there is no clear theory or data, one needs to simply choose a value, but then one needs to show, what the sensitivity of the results (and conclusions) to this choice is. I suggest a sensitivity study, where the number of regions is being gradually changed, maybe they find a domain where the calculated representativeness is relatively stable?

A3

The effect of the total number of ecoregions on the analysis is actually quite low; This seems to have been described inadequately in the previous manuscript version, therefore we will aim to better communicate this.

The actual network representativeness is calculated on a pixel scale, not on an ecoregion scale. Ecoregions were not involved in this step. For visualization purposes, and with the aim to contextualize our results through the ER1 and ER5 (now: ER4) metrics, we used ecoregions. We did show the raw representativeness values in table 2 and figure 6, but again these results were unaffected by the ecoregions.

The choice of ecoregions has minor effects on the preselection of potential new observation sites, since here we removed potential sites that are present in ecoregions that already have a site present. With a low number of (large) ecoregions, the number of sites that would be removed would be quite high, which would not be useful for our optimization attempts. Similarly, with a high number of small ecoregions the number of sites would not be reduced enough. Line 419-425 describes the effect of duplicate sites in ecoregions on the optimization with only 0.1 to 1% improvement globally. Accordingly, the choice of the total number of ecoregions does not hold the potential to significantly change the output of our representativeness analyses.

To make it clearer where the ecoregion settings are influencing our analyses, we plan to divide the previous paragraph 2.2 into two new sections 2.2 and 2.3. 2.2 will describe the representativeness assessment as it is now, whereas 2.3 will be called quantification of results.

In the introduction we add the following line at the end of the second but last paragraph: *"In our implementation we will also perform this analysis on an individual pixel scale."* Moreover, in line 155 we add the considered k values: "*(35 100 200 500 1000)*".

- If the authors agree that there are still large uncertainties in the outcomes from their study, I would like to ask them for presenting this message clearly in the text: a section in the discussion challenges for extrapolation to the artic through sample sparseness and prospects for continental and global extrapolation from eddy covariance flux tower networks in the Arctic and probably the globe, too. This uncertainty should also be mentioned as a main result from the study, in the conclusions and the abstract /and even in the title, if possible).

A4

We agree with the reviewer that the uncertainties associated with our extrapolation approach should be placed more prominently in the manuscript text. The changes listed in answers A1-A3 above should address a large part of these uncertainties, and we agree that some of the uncertainties can never be removed when dealing with studies of this nature. More emphasis can be given to upscaling uncertainties, and we believe the new ER4 metric actually facilitates this.

We plan to change the central abstracts paragraph to:

*"This study provides an inventory of Arctic (here $>= 60^O$N) EC sites, which has also been made available online (https://cosima.nceas.ucsb.edu/carbon-flux-sites/). Our database currently comprises 120 EC sites, but only 83 are listed as active, and just 25 of these active sites remain operational throughout the winter. To map the representativeness of this EC network, we*

*evaluated the similarity between environmental conditions observed at the tower locations and those within the larger Arctic study domain based on 18 bioclimatic and edaphic variables. This allows us to assess a general level of similarity between ecosystem conditions within the domain, but does not reflect changes in greenhouse gas flux rates directly. We define two metrics based on this representativeness score, one that measures whether a location is represented by an EC tower with similar characteristics (ER1) and one where we assess if a minimum level of representation for statistically rigorous extrapolation is met (ER4). We find that while half of the domain is represented by at least one tower, only a third has enough towers in similar locations to allow reliable extrapolation. When we consider methane measurements or year-round (including wintertime) measurements the values drop to about one fifth and one tenth of the domain, respectively. With the majority of sites located in Fennoscandia and Alaska, these regions were assigned the highest level of network representativeness, while large parts of Siberia and patches of Canada were classified as under-represented. Across the Arctic, particularly mountainous regions were poorly represented by the current EC observation network."*

To further highlight these uncertainties, we propose the following addition to the discussion section:

*"A focus on the ER4 metric indicates that only one third of the Arctic can be represented with a high statistical power (Table 2, All, 5 year, Active), and if we consider the wintertime networks as the configurations with the only reliable year round carbon budget, this value drops to about a tenth of the Arctic domain. This constitutes an important gap in data coverage, since while wintertime fluxes in the Arctic are substantially lower than those during summer time, they are still significant for Arctic carbon budgets (Zimov et al., 1996; Wille et al., 2008; Euskirchen et al., 2012; Marushchak et al., 2013; Lüers et al., 2014; Oechel et al., 2014; Zona et al., 2016; Natali et al., 2019)."*

Finally, we will make the following changes to the Conclusion section that also highlight better the uncertainties associated with our extrapolation approach:

*"The Arctic is warming and changing rapidly, with implications both for global climate change trajectories and the livelihood of local communities. Large investments into adequate research infrastructure is required in the future to improve understanding of these Arctic changes across all relevant scales, and support the development of mitigation and adaptation measures. To efficiently use resources for an optimum upgrade of observational facilities, we need to advance our understanding on what our current measurements represent, and where gaps remain.*

*This study helps to guide efficient upgrades of the Arctic greenhouse gas monitoring facilities, showing that even though the Arctic EC network has grown considerably over the past decades, only half the Arctic is represented by an EC tower at all, and this value drops to one third of the domain when we consider a statistically rigorous number of EC towers for upscaling. In particular, coverage within Siberia, Canada and mountainous regions is lacking. There are also large gaps when it comes to year-round data coverage, and non-CO2 fluxes, with less than 20% of the Arctic terrestrial domain currently being covered by these measurements. While these numbers are associated with considerable uncertainties since we do not directly quantify how*

*differences in ecosystem characteristics translate to fluxes, the applicability of this approach has been demonstrated by numerous previous extrapolation studies using similar underlying data as their explanatory variables. Accordingly, data-driven upscaling of EC databases to produce pan-Arctic greenhouse gas budgets, training datasets for biosphere process models or prior flux fields for atmospheric inverse modelling are still associated with large uncertainties, given the size of the regions currently underrepresented. We propose and test several methods for optimizing the EC network based on this representativeness assessment, and provide recommendations on network upgrades based on the best-performing and most practical option. Overall, as the most cost-efficient strategy for network improvements, we recommend upgrading selected existing locations with new instrumentation for methane measurements, since large coverage gaps for this important greenhouse gas currently severely compromise our ability to comprehensively monitor carbon release from degrading permafrost within the extensive Arctic landscapes. Furthermore, keeping sites operational during the winter has been shown to be essential to understand annual carbon budgets within the Arctic, and also in this context winter-proofing strategically selected existing sites would provide the most efficient pathway towards better network coverage. A final step would be to extend the network further, especially in Siberia and Canada, and our method can help with selecting those locations that improve overall network coverage best.*

*This study, and associated datasets, has been designed to help the Arctic research community in planning future Arctic EC stations, and improve the quantification of uncertainties in the context of upscaling activities. Future studies could expand upon this study by selecting hard to sample regions, such as ecoregions without any current sampling and without villages or infrastructure, as a target for temporary towers or flight campaigns to empirically assess their (dis)similarity to already sampled ecoregions. Seasonal campaigns in mountainous regions could verify the assumption that fluxes in high latitude high elevation regions are low enough not to warrant the high investment and operation costs of permanent towers there. An assessment of heterogeneity in domains where replicative EC studies have been performed might provide guidance in better quantifying ecoregions sizes in data space for representativeness comparison."*

I do not say that this approach isn't useful, but, honestly, I wonder, what the scientific progress is beyond some practical value. A clear prospect from this study is that it provides guidance (even best possible based on available information) for the choice of locations of additional sites and especially the evaluation of alternative choices between establishing new sites versus upgrading existing sites. But the guidance rests on untested assumptions or hypotheses that cannot be tested. I can understand, why the authors do what they do, and they have certainly chosen one of the best possible and most careful approaches, but whether this is good enough to guide investment into research remains an open question. In order to not leave readers with unrealistic expectations, this risk should be discussed and mentioned at all visible places, including the abstract and the conclusions.

See previous answer (A4)

Detailed comments:

Abstract

24-26: Can you briefly add representativeness for what – for the driver field that you have, all same important, or for the results from the network (which, CO2, GHG, energy radiation etc.). Later you write "capture the spatio-temporal variability of surface atmosphere exchange fluxes across the pan-Arctic ecosystem distribution." (lines 77-78)

24-26: we plan to change the sentence to:

To map the representativeness of this EC network, we evaluated the similarity between environmental conditions observed at the tower locations and those within the larger Arctic study domain based on 18 bioclimatic and edaphic variables.

77-78 we indeed do not calculate flux variability directly. With rewriting this paragraph (see A1), this will be taken into account.

In general: Address the limitation of the approach that it is based on the unproven assumption and why it is still the best that we can do.

See our answer A1: the method is not unproven, there are numerous references that have applied it earlier. While a full explanation of the limitations has to wait until the methods and discussion section we can add here:

"This allows us to assess a general level of similarity between pixels within the domain, but does not directly reflect changes in greenhouse gas flux rates."

Introduction

39 please try to find a more accurate term than "vast"

We plan to change vast stocks of carbon to "*large pools of soil organic carbon*"

64-67: Please elaborate either here or in the discussion, which conditions must be fulfilled for a site being representative for a complete region or class of ecosystems and how the coarseness of the classification system affects this representativeness (see section 4.3).

See our answer A2 for details. We will change the last paragraph of 2.2 (now 2.3) to introduce ER4, and make clear that ER1 is not statistical sufficient for upscaling.

70 Please add "high precision concentration" in front of "tall tower networks" and note the fundamental difference in the approaches. BTW Scientifically, the definition of representativeness is much more straight forward for an atmospheric concentration network (concentration footprint coverage, improvement of posterior probability of fluxes estimated with model inversion) than for an ecosystem flux network with much more local representation. Contrasting these two approaches would be a very good example to illustrate the challenge with (and the limitations of) the upscaling of flux data to the large spatial scale.

We plan to add *"high precision concentration"*.

The difference between the two is an interesting point; however, we believe that discussion does not fit the purpose of this paragraph.

To acknowledge the difference, we plan to add: *".. which can be utilized to integrate fluxes on regional scales through their large footprints, to the .."*

79-80 "We quantify representativeness here based on the similarity in key ecosystem characteristics of any location in our domain to those of the EC sites." without showing that this prior assumption is true. This is a key limitation. to alleviate this, please provide evidence (or at least theory) that this approach is (likely) valid and test it with the data available (e.g. measured GHG fluxes in ecoregions are statistically different from measured GHG fluxes in contrasting ecoregions).

See answer A1

89-81 "This concept is similar to producing gridded products by upscaling localized flux data to a larger region". Please explain to make sure what is similar and what is different between your approach and the common upscaling approaches. It is not like Kriging or other geostatistical approaches. A mathematically rigorous geostatistical approach would, e.g., relate the quantitative relationship between the site proxies (i.e. your 18 variables) and the fluxes (e.g. confidence band for prediction) and then estimate the upscaling error in bootstrapping approaches etc. . If I am right, the proposed approach lacks this mathematical rigor. Please comment.

We plan to rewrite this paragraph (see A1). We will focus on the method itself instead of any similarity to upscaling.

136-139 Please list and define the variables here and describe for each variable the relevance for determining the GHG fluxes. Then argue for the choice of so many variables, why 18? A sensitivity study for the choice of the variables (importance of the variables for establishing a stable spatial ecoregion patterns) could, e.g., support the choice.

We plan to add a section to the appendix where we go in more detail on the variables, see answer A1.

141 Give the pixel size (1 km^2, if the same as in the reference)

Indeed. We will add this information to the text.

BTW- in Tab. A1: GlobPermafrost doi: 10.1111/j.1461-0248.2004.00694.x and Saxon et. al. 2005 doi: 10.1111/j.1461-0248.2004.00694.x have the same DOI, is that an error?

This was an error and will be updated to the proper doi *10.1594/PANGAEA.888600*.

146-164: In this reasoning the choice of k=100 is subjective. Translating the "gut feeling" terms of "truly coherent" or "would not grant much improvement" into clear scientific (statistical) concepts would alleviate this limitation. I am not an expert, but wouldn't a sensitivity study to relate (small) changes in K to the possible existence of a stable frequency distribution of representativeness values be of some help here? The interpretation would be the resolution matching the scale of natural variability.

See our answer A3 above for details. While there is some subjectivity involved in our choices here, since the ecoregions were not directly involved in the calculation of the representativeness values there was no clear metric that would lend itself to statistical test. Also, there is no clear definition of what the size or variability of an ecosystem or ecoregion must be, thus as we state in the text: we selected k=100 as a compromise between ecosystem coherence and representativeness that agrees well with our study objectives.

159-163 A very good move to compare the results to independent information. Later I suggest a small results section where you can give some statistical similarity figures between the ecoregions and CAVM. For interpretation of the comparison, please add to which degree the definition of CAVM units uses similar or different information compared to the proposed approach.

Results can be found in section 3.2 from line 369

The following additional information will be added to compare the two methods:

*"The key differences between our method and the CAVM rasterized maps is that for the latter clustering was done on sub regions of the original CAVM map using AVHRR and MODIS (red and infrared channels, and NDVI) as well as elevation data from the Digital Chart of the World. They then aggregated the clustering units to their CAVM vegetation units using a wide range of auxiliary data such as regional vegetation maps, ground-based studies as well as GoogleEarth imagery."*

164-168 and following down to 178. Please, see my point 2 above. Note that requiring a minimum of 5 sites is a subjective choice and should be substantiated by statistical uncertainty parameters. Please specify the quality requirements for the 5 stations (CH4 fluxes, wintertime, length of observation period, etc.).

See our answer A2 above: the now 4 stations are substantiated by statistical power analysis.

2.4 Many of these choices make a lot of sense to me, especially including the station quality and consequently contrasting upgrading versus new locations. For the role of the random sampling versus guided sampling (262-269), I wonder whether the improvement of the guided sampling over the random sample isn't a bit of a circular argument, as the output of the analysis is the consequence of the approach and the initial assumptions. But the degree, on how the difference between those two sampling modes are and how they develop with increasing samples, characterizes the approach and also the relationships in the data. Maybe one could help the reader by explaining this more explicitly.

We plan to add the following text to clarify this method:
*"The guided approach should see large gains in initial network development, as the most optimal sites are chosen first. Consequently, with the best locations already been selected, later additions will have a reduced impact on the network representativeness. Using a random site selection method, we expect initial improvements to be lower, but at the same time the decline in improvement per additional site will be less since later additions might still contain high impact locations. While normally sites are not strictly selected at random, they are typically not chosen with the entire network in mind, and some opportunism exists as far as accessibility, funding, and existing infrastructure."*

Results

The section 3.1. is very well especially the combination of Figure 2 presents the temporal development of the data quality and quantity in a very intuitive way.

Thanks for appreciating this.

280 Figure 1: Although to me the high resolution map is esthetically very appealing, wouldn't a map with ecoregions as a background (same colour code as stations) include more useful information in this context?

Since the ecoregions are only a small part of the analysis (see also A3), we prefer to keep this map in the current format, but we can add the ecoregions map to the appendix.

326: "excellent data coverage", this raises a lot of good feelings, but isn't it just the "highest data coverage"? Is it also excellent? if yes, why?

Agreed, we plan to change this to *"highest data coverage"*.

Between section 3.1 and 3.2 I miss a section that presents the geostatistical results, e.g. a map of ecoregions, maybe including the comparison with the CAVM distribution (if you consider this, move 268-371 to this subsection).

We do not plan a geostatistical analysis (last answer A2), and are of the opinion that the ecoregions are not impactful enough to warrant an entire section (A3). A comparison map between CAVM and our aggregated ecoregions can be found in Appendix B. We can add the ecoregions map to the Appendix, if desired.

Section 3.2 does a very good job to present the strengths of the approach, i.e. providing a spatial data base that is an excellent basis for both spatially resolved presentation and for calculating summary statistics. Much of what is presented is a logical consequence of the approach (e.g. the differences between ER1 and ER2 standards) but the concrete application to the data sets brings about some interesting features and quantifications that turn out to have some plausible possible causes

Thank you for highlighting this.

345 it is interesting, that elevation (and topography) are not directly part of the site factor variables. It looks, as if this would be good and commonly available candidates.

Elevation (and topography) are part of the *Compound Topographic Index* variable. The planned additions to appendix 1 will explain this. (A1)

373 replace "representativeness" by "representativeness"

We plan to correct this typo.

Section 3.3. is very interesting to read. You have tested a lot of plausible settings and compared the results, which gives the impression that, despite the different possibilities, the main results where to place new stations and where to upgrade them are relatively uniform. This qualifies the robustness of your approach! (But is does not necessarily qualify the study to be robust, because of the hence unavoidable risk of systematic errors that an optimized network design improves the representativeness, but still does not achieve the level needed for an sufficiently accurate flux extrapolation to the Arctic. )

We agree on the statement regarding ER1 and ER5 metric results, The introduction of the ER4 metric based on the work by Hill et al. (2017) should give us an indication of the level required for sufficient accurate flux extrapolation.

398: Fig. 5:

- please explain what "pre optimized" means. It doesn't show in the text.
- Mention the number of added stations are (n = 5, if I am right) and that the location of these new stations are optimized regarding the parameter of interest.
- How would the selection look like, if one optimizes for representativeness of all combined, maybe, weighted to optimally represent the global warming effect?

Otherwise fun to see, how higher representativeness areas develop around the newly proposed stations.

We will change the first sentence of section 3.3 to: "*The targeted selection of new site locations to sequentially fill the biggest gaps in the existing network, as executed by these stepwise optimization approaches, yields clear improvements in overall network coverage (Fig. 5) compared to the conditions before the optimization (i.e. the 'current' network as shown in figures 3,4, and table 2)."*

And indeed we should mention that 5 sites have been added: "*Network improvement compared to original coverage after adding 5 sites with the targeted selection."*

Regarding the third item, this is good question indeed, one we contemplated ourselves earlier, but decided that this topic is beyond the scope of the presented material. Weighting by global warming effects is a good choice, but is a bit complicated to implement with regards to the winter fluxes. We could use the few existing studies to get an indication of the fraction of the

winter fluxes compared to the entire year; however, to make the comparison balanced, we should also weigh for the cost of each extension or upgrade. This is technically doable, and something we want to revisit in the future. But considering the different network configurations, optimization methods, quantification metrics we were of the opinion the paper would rather suffer from adding even more results.

What we can do now is compare the optimization of *winter*, *methane* and *winter-methane* configurations with that of the *all* network. In this context, we find at least two ecoregions that are chosen in the top 10 of all 4 configurations, which indicates there are locations that would support data coverage across different scenarios.

450 replace "manuscript" by "this study"

We will change this.

467 reconsider using 'higher' (relative term) instead of 'good' (absolute term) coverage, as it is not sure for what the coverage is sufficiently good.

We plan to change it to: *"sufficient coverage according to the ER4 metric"*

482-497 Here you discuss rightly the limitations of your study, make sure that this is explained at the most visible parts of the manuscript (conclusions and abstract), too.

Agreed, please see other answers above.

484 -486: Please show, how "the maps can be utilized as a measure representing the extrapolation uncertainty" … . Given what is written from lines 492 including "specific fluxes provided by the eddy covariance tower network based on these data layers must largely remain qualitative, since no clear quantitative linkage between the bioclimatic controls and the fluxes for CO2 and CH4 can be considered.", I would even doubt it.

489 either use "a priori" or "prior"

Agreed, plan to change to: *"a priori"*

Section 4.3 The role of small scale variability is a very important, if not crucial aspect, for your study, because it questions one major assumption to this approach, i.e. that data from one site does represent data from other sites and even entire regions. Please also note that you define the meaning of subscale variability that is unaccounted for in the land-cover classification by the choice of the k value. As long as there is no clear assessment on how many towers are subject to small scale effects, the representativeness value of the tower network is highly speculative. I would also include here the length of the observation period and whether or not winter fluxes and CH4 fluxes are being measured. The required minimum length of observation periods should be estimated from typical time scales of interannual variability. Please make sure that readers get this information on the very substantial and serious challenges of upscaling from tower networks to large surface units.

The first part of this comment will be addressed by the updated ER1 and ER4 metrics. We fully agree that the length of observation period is an important factor, especially when taking interanual variability and wintertime measurements into account. This is why we differentiated also the *5 year active* and *wintertime* active subsets. A more detailed assessment of the effect of temporal variability in data coverage, however, will be performed in upcoming work. For now, we plan to add the following statement to the manuscript text to clarify the importance of observation length (Section 4.1), since here we already discuss the networks activity and wintertime gaps.

*"We would like to highlight that while the main focus of our analysis was on the spatial pattern of measurements, the temporal distribution of measurements is an important aspect as well, and not all temporal effects could be captured by the method applied for this study. On a short temporal scale, data gaps are a problem that needs to be resolved when calculating annual budgets. A typical tower has a data coverage of 65% (Falge et al., 2001), and considering the typical winter time shutdown and more extreme weather, this value will be lower in the Arctic. And while there are gap filling methods, the errors of these methods increases with gap size (Falge et al., 2001; Moffat et al., 2007). Furthermore, it should be noted that long time series are exceptionally valuable for studying ecosystem feedbacks with climate, as explained e.g. by Baldocchi (2020), and especially interannual variability and long term ecological trends are impossible to detect without long term observations."*

527: Please explain, why can you assume something, when the information to substantiate any possible assumption is simply lacking? If I am right, such an assumption has no scientific value. Rather discuss which alternatives exist and what the lack of information means to the risk of drawing false conclusions from your results.

We will now instead highlight this limitation and indicate (future) options to better address this: *"We are aware of this problem, but lack the database to quantify it at this time. For some locations in this study, there are multiple towers within the same pixel, therefore we can capture the effect of small scale variability. Maost towers, however, are the sole measurements in their pixel. Over the coming decades, gridded products based on satellite observations are expected to increase in availability, and also their spatial resolution will improve. This can grant opportunities in the future to look at current sub pixel heterogeneity, or simply assess variability at such a small scale that subpixel heterogeneity is no longer a serious concern. "*

sub-section 4.4

good and practically important – no other comments

Thanks

Conclusions

Please limit to conclusions on novel and important results and interpretations from your study. Most of what is said does not address these. The underrepresentation of Siberia and Canada was

already noticed in other publications. I miss anything on consequences from the strengths and limitations of the proposed new approach and, equally important, what it means for the (humbling) prospects on extrapolating data from 120 flux sites to a global region.

We will specify our novel results in more detail and put more emphasis on the limitations of this study and upscaling limitations from this network. See also our response A4.